# IMPROVING TRANSFORMATION INVARIANCE IN CONTRASTIVE REPRESENTATION LEARNING

**Adam Foster**[*]**, Rattana Pukdee**[*]**& Tom Rainforth**
Department of Statistics
University of Oxford
`{adam.foster,rainforth}@stats.ox.ac.uk`

## ABSTRACT

We propose methods to strengthen the invariance properties of representations obtained by contrastive learning. While existing approaches implicitly induce a degree of invariance as representations are learned, we look to more directly enforce invariance in the encoding process. To this end, we first introduce a training objective for contrastive learning that uses a novel regularizer to control how the representation changes under transformation. We show that representations trained with this objective perform better on downstream tasks and are more robust to the introduction of nuisance transformations at test time. Second, we propose a change to how test time representations are generated by introducing a feature averaging approach that combines encodings from multiple transformations of the original input, finding that this leads to across the board performance gains. Finally, we introduce the novel Spirograph dataset to explore our ideas in the context of a differentiable generative process with multiple downstream tasks, showing that our techniques for learning invariance are highly beneficial.

## 1 INTRODUCTION

Learning meaningful representations of data is a central endeavour in artificial intelligence. Such representations should retain important information about the original input whilst using fewer bits to store it (van der Maaten et al., 2009; Gregor et al., 2016). Semantically meaningful representations may discard a great deal of information about the input, whilst capturing what is relevant. Knowing what to discard, as well as what to keep, is key to obtaining powerful representations.

By defining transformations that are believed *a priori* to distort the original without altering semantic features of interest, we can learn representations that are (approximately) invariant to these transformations (Hadsell et al., 2006). Such representations may be more efficient and more generalizable than lossless encodings. Whilst less effective for reconstruction, these representations are useful in many downstream tasks that relate only to the semantic features of the input. Representation invariance is also a critically important task in of itself: it can lead to improved robustness and remove noise (Du et al., 2020), afford fairness in downstream predictions (Jaiswal et al., 2020), and enhance interpretability (Xu et al., 2018).

Contrastive learning is a recent and highly successful self-supervised approach to representation learning that has achieved state-of-the-art performance in tasks that rely on semantic features, rather than exact reconstruction (van den Oord et al., 2018; Hjelm et al., 2018; Bachman et al., 2019; He et al., 2019). These methods learn to match two different transformations of the same object in representation space, distinguishing them from contrasts that are representations of other objects.

The objective functions used for contrastive learning encourage representations to remain similar under transformation, whilst simultaneously requiring different inputs to be well spread out in representation space (Wang & Isola, 2020). As such, the choice of transformations is key to their success (Chen et al., 2020a). Typical choices include random cropping and colour distortion.

However, representations are compared using a similarity function that can be maximized even for representations that are far apart, meaning that the invariance learned is relatively weak. Unfor-

---

[*]Equal contribution

tunately, directly changing the similarity measure hampers the algorithm (Wu et al., 2018; Chen et al., 2020a). We therefore investigate methods to improve contrastive representations by explicitly encouraging stronger invariance to the set of transformations, without changing the core self-supervised objective; we look to extract more information about how representations are changing with respect to transformation, and use this to direct the encoder towards greater invariance.

To this end, we first develop a gradient regularization term that, when included in the training loss, forces the encoder to learn a representation function that varies slowly with continuous transformations. This can be seen as *constraining* the encoder to be approximately transformation invariant. We demonstrate empirically that while the parameters of the transformation can be recovered from standard contrastive learning representations using just linear regression, this is no longer the case when our regularization is used. Moreover, our representations perform better on downstream tasks and are robust to the introduction of nuisance transformations at test time.

Test representations are conventionally produced using untransformed inputs (Hjelm et al., 2018; Kolesnikov et al., 2019), but this fails to combine information from different transformations and views of the object, or to emulate settings in which transformation noise cannot simply be removed at test time. Our second key proposal is to instead create test time representations by feature averaging over multiple, differently transformed, inputs to address these concerns and to more directly impose invariance. We show theoretically that this leads to improved performance under linear evaluation protocols, further confirming this result empirically.

We evaluate our approaches first on CIFAR-10 and CIFAR-100 (Krizhevsky et al., 2009), using transformations appropriate to natural images and evaluating on a downstream classification task. To validate that our ideas transfer to other settings, and to use our gradient regularizer within a fully differentiable generative process, we further introduce a new synthetic dataset called Spirograph. This provides a greater variety of downstream regression tasks, and allows us to explore the interplay between nuisance transformations and generative factors of interest. We confirm that using our regularizer during training and our feature averaging at test time both improve performance in terms of transformation invariance, downstream tasks, and robustness to train–test distributional shift.

In summary, the contributions of this paper are as follows:

- We derive a novel contrastive learning objective that leads to more invariant representations.
- We propose test time feature averaging to enforce further invariance.
- We introduce the Spirograph dataset.
- We show empirically that our approaches lead to more invariant representations and achieve state-of-the-art performance for existing downstream task benchmarks.

## 2    PROBABILISTIC FORMULATION OF CONTRASTIVE LEARNING

The goal of unsupervised representation learning is to encode high-dimensional data, such as images, retaining information that may be pertinent to downstream tasks and discarding information that is not. To formalize this, we consider a data distribution $p(\mathbf{x})$ on $\mathcal{X}$ and an encoder $\mathbf{f}_\theta : \mathcal{X} \to \mathcal{Z}$ which is a parametrized function mapping from data space to representation space.

Contrastive learning is a self-supervised approach to representation learning that learns to make representations of differently transformed versions of the same input more similar than representations of other inputs. Of central importance is the set of transformations, also called augmentations (Chen et al., 2020a) or views (Tian et al., 2019), used to distort the data input $\mathbf{x}$. In the common application of computer vision, it is typical to include resized cropping; brightness, contrast, saturation and hue distortion; greyscale conversion; and horizontal flipping. We will later introduce the Spirograph dataset which uses quite different transformations. In general, transformations are assumed to change the input only cosmetically, so all semantic features such as the class label are preserved; the set of transformations indicates changes which can be safely ignored by the encoder.

Formally, we consider a transformation set $\mathcal{T} \subseteq \{\mathbf{t} : \mathcal{X} \to \mathcal{X}\}$ and a probability distribution $p(\mathbf{t})$ on this set. A representation $\mathbf{z}$ of $\mathbf{x}$ is obtained by applying a random transformation $\mathbf{t}$ to $\mathbf{x}$ and then encoding the result using $\mathbf{f}_\theta$. Therefore, we do not have one representation of $\mathbf{x}$, but an implicit distribution $p(\mathbf{z}|\mathbf{x})$. A sample of $p(\mathbf{z}|\mathbf{x})$ is obtained by sampling $\mathbf{t} \sim p(\mathbf{t})$ and setting $\mathbf{z} = \mathbf{f}_\theta(\mathbf{t}(\mathbf{x}))$.

If the encoder is to discard irrelevant information, we would expect different encodings of $\mathbf{x}$ formed with different transformations $\mathbf{t}$ to be close in representation space. Altering the transformation should not lead to big changes in the representations of the same input. In other words, the distribution $p(\mathbf{z}|\mathbf{x})$ should place most probability mass in a small region. However, this does not provide a sufficient training signal for the encoder $\mathbf{f}_\theta$ as it fails to penalize trivial solutions in which all $\mathbf{x}$ are mapped to the same $\mathbf{z}$. To preserve meaningful information about the input $\mathbf{x}$ whilst discarding purely cosmetic features, we should require $p(\mathbf{z}|\mathbf{x})$ to be focused around a single $\mathbf{z}$ whilst *simultaneously* requiring the representations of different inputs not to be close. That is, the marginal $p(\mathbf{z}) = \mathbb{E}_{p(\mathbf{x})}[p(\mathbf{z}|\mathbf{x})]$ should distribute probability mass over representation space.

This intuition is directly reflected in contrastive learning. Most state-of-the-art contrastive learning methods utilize the InfoNCE objective (van den Oord et al., 2018), or close variants of it (Chen et al., 2020a). InfoNCE uses a batch $\mathbf{x}_1, ..., \mathbf{x}_K$ of inputs, from which we form pairs of representations $(\mathbf{z}_1, \mathbf{z}_1'), ..., (\mathbf{z}_K, \mathbf{z}_K')$ by applying two random transformations to each input followed by the encoder $\mathbf{f}_\theta$. In probabilistic language

$$\mathbf{x}_i \sim p(\mathbf{x}) \text{ for } i = 1, ..., K \tag{1}$$

$$\mathbf{z}_i, \mathbf{z}_i' \sim p(\mathbf{z}|\mathbf{x} = \mathbf{x}_i) \text{ conditionally independently given } \mathbf{x}_i, \text{ for } i = 1, ..., K, \tag{2}$$

such that $\mathbf{z}_i, \mathbf{z}_i' = \mathbf{f}_\theta(\mathbf{t}(\mathbf{x})), \mathbf{f}_\theta(\mathbf{t}'(\mathbf{x}))$ for i.i.d. transformations $\mathbf{t}, \mathbf{t}' \sim p(\mathbf{t})$. Given a learnable similarity score $s_\phi : \mathcal{Z} \times \mathcal{Z} \to \mathbb{R}$, contrastive learning methods minimize the following loss

$$\mathcal{L}(\theta, \phi) = -\frac{1}{K}\sum_{i=1}^{K} s_\phi(\mathbf{z}_i, \mathbf{z}_i') + \frac{1}{K}\sum_{i=1}^{K}\log\left(\sum_{j=1}^{K}\exp\left[s_\phi(\mathbf{z}_i, \mathbf{z}_j')\right]\right). \tag{3}$$

Written in this way, we see that the loss will be minimized when $s_\phi(\mathbf{z}_i, \mathbf{z}_i')$ is large, but $s_\phi(\mathbf{z}_i, \mathbf{z}_j')$ is small for $i \neq j$. In other words, InfoNCE makes the two samples $\mathbf{z}_i, \mathbf{z}_i'$ of $p(\mathbf{z}|\mathbf{x} = \mathbf{x}_i)$ similar, whilst making samples $\mathbf{z}_i, \mathbf{z}_j'$ of $p(\mathbf{z})$ dissimilar. This can also be understood through the lens of mutual information, for more details see Appendix A.

In practice, the similarity measure used generally takes the form (Chen et al., 2020a)

$$s_\phi(\mathbf{z}, \mathbf{z}') = \frac{\boldsymbol{g}_\phi(\mathbf{z})^\top \boldsymbol{g}_\phi(\mathbf{z}')}{\tau\|\boldsymbol{g}_\phi(\mathbf{z})\|_2\|\boldsymbol{g}_\phi(\mathbf{z}')\|_2} \tag{4}$$

where $\boldsymbol{g}_\phi$ is a small neural network and $\tau$ is a temperature hyperparameter. If the encoder $\mathbf{f}_\theta$ is perfectly invariant to the transformations, then $\mathbf{z}_i = \mathbf{z}_i'$ and $s_\phi(\mathbf{z}_i, \mathbf{z}_i')$ will be maximal. However, there are many ways to maximize the InfoNCE objective without encouraging strong invariance in the encoder.[1] In this paper, we show how we can learn stronger invariances, above and beyond what is learned through the above approach, and that this benefits downstream task performance.

## 3 INVARIANCE BY GRADIENT REGULARIZATION

Contrastive learning with InfoNCE can gently encourage invariance by maximizing $s_\phi(\mathbf{z}, \mathbf{z}')$, but does not provide a strong signal to *ensure* this invariance. Our first core contribution is to show how we can use gradient methods to directly regulate how the representation changes with the transformation and thus ensure the desired invariance. The key underlying idea is to **differentiate the representation with respect to the transformation**, and then encourage this gradient to be small so that the representation changes slowly as the transformation is varied.

To formalize this, we begin by looking more closely at the transformations $\mathcal{T}$ which are used to define the distribution $p(\mathbf{z}|\mathbf{x})$. Many transformations, such as brightness adjustment, are controlled by a *transformation parameter*. We can include these parameters in our set-up by writing the transformation $\mathbf{t}$ as a map from both input space $\mathcal{X}$ and transformation parameter space $\mathcal{U}$, i.e. $\mathbf{t} : \mathcal{X} \times \mathcal{U} \to \mathcal{X}$. In this formulation, we sample a random transformation parameter from $\mathbf{u} \sim p(\mathbf{u})$ which is a distribution on $\mathcal{U}$. A sample from $p(\mathbf{z}|\mathbf{x})$ is then obtained by taking $\mathbf{z} = \mathbf{f}_\theta(\mathbf{t}(\mathbf{x}, \mathbf{u}))$, with $\mathbf{t}$ now regarded as a fixed function.

---

[1]This is because the function $\boldsymbol{g}_\phi$ is not an injection, so we may have $\boldsymbol{g}_\phi(\mathbf{z}) = \boldsymbol{g}_\phi(\mathbf{z}')$ but $\mathbf{z} \neq \mathbf{z}'$. Johnson & Lindenstrauss (1984) gives conditions under which a projection of this form will preserve approximate distances, in particular, the required projection dimension is much larger than the typical value 128.

The advantage of this change of perspective is that it opens up additional ways to learn stronger invariance of the encoder. In particular, it may make sense to consider the gradient $\nabla_{\mathbf{u}}\mathbf{z}$, which describes the rate of change of $\mathbf{z}$ with respect to the transformation. This only makes sense for some transformation parameters—we can differentiate with respect to the brightness scaling but not with respect to a horizontal flip.

To separate out differentiable and non-differentiable parameters we write $\mathbf{u} = \boldsymbol{\alpha}, \boldsymbol{\beta}$ where $\boldsymbol{\alpha}$ are the parameters for which it makes sense to consider the derivative $\nabla_{\boldsymbol{\alpha}}\mathbf{z}$. Intuitively, this gradient should be small to ensure that representations change only slowly as the transformation parameter $\boldsymbol{\alpha}$ is varied. For clarity of exposition, and for implementation practicalities, it is important to consider gradients of a *scalar* function, so we introduce an arbitrary direction vector $\mathbf{e} \in \mathcal{Z}$ and define

$$F(\boldsymbol{\alpha},\ \boldsymbol{\beta},\ \mathbf{x},\ \mathbf{e}) = \mathbf{e} \cdot \frac{\mathbf{f}_\theta(\mathbf{t}(\mathbf{x},\boldsymbol{\alpha},\boldsymbol{\beta}))}{\|\mathbf{f}_\theta(\mathbf{t}(\mathbf{x},\boldsymbol{\alpha},\boldsymbol{\beta}))\|_2} \tag{5}$$

so that $F : \mathcal{A} \times \mathcal{B} \times \mathcal{X} \times \mathcal{Z} \to \mathbb{R}$ calculates the scalar projection of the normalized representation $\mathbf{z}/\|\mathbf{z}\|_2$ in the $\mathbf{e}$ direction. To encourage an encoder that is invariant to changes in $\boldsymbol{\alpha}$, we would like to minimize the *expected conditional variance* of $F$ with respect to $\boldsymbol{\alpha}$:

$$V = \mathbb{E}_{p(\mathbf{x})p(\boldsymbol{\beta})p(\mathbf{e})} \left[ \mathrm{Var}_{p(\boldsymbol{\alpha})}[F(\boldsymbol{\alpha},\boldsymbol{\beta},\mathbf{x},\mathbf{e}) \mid \mathbf{x},\boldsymbol{\beta},\mathbf{e}] \right], \tag{6}$$

where we have exploited independence to write $p(\mathbf{x},\boldsymbol{\beta},\mathbf{e}) = p(\mathbf{x})p(\boldsymbol{\beta})p(\mathbf{e})$. Defining $V$ requires a distribution for $\mathbf{e}$ to be specified. For this, we make components of $\mathbf{e}$ independent Rademacher random variables, justification for which is included in Appendix B.

A naive estimator of $V$ can be formed using a direct nested Monte Carlo estimator (Rainforth et al., 2018) of sample variances, which, including Bessel's correction, is given by

$$V \approx \frac{1}{K} \sum_{i=1}^{K} \left( \frac{1}{L-1} \sum_{j=1}^{L} F(\boldsymbol{\alpha}_{ij},\boldsymbol{\beta}_i,\mathbf{x}_i,\mathbf{e}_i)^2 - \frac{1}{L(L-1)} \left[ \sum_{k=1}^{L} F(\boldsymbol{\alpha}_{ik},\boldsymbol{\beta}_i,\mathbf{x}_i,\mathbf{e}_i) \right]^2 \right) \tag{7}$$

where $\mathbf{x}_i,\boldsymbol{\beta}_i,\mathbf{e}_i \sim p(\mathbf{x})p(\boldsymbol{\beta})p(\mathbf{e})$ and $\boldsymbol{\alpha}_{ij} \sim p(\boldsymbol{\alpha})$. However, this estimator requires $LK$ forward passes through the encoder $\mathbf{f}_\theta$ to evaluate. As an alternative to this computationally prohibitive approach, we consider a first-order approximation[2] to $F$

$$F(\boldsymbol{\alpha}',\boldsymbol{\beta},\mathbf{x},\mathbf{e}) - F(\boldsymbol{\alpha},\boldsymbol{\beta},\mathbf{x},\mathbf{e}) = \nabla_{\boldsymbol{\alpha}}F(\boldsymbol{\alpha},\boldsymbol{\beta},\mathbf{x},\mathbf{e}) \cdot (\boldsymbol{\alpha}' - \boldsymbol{\alpha}) + o(\|\boldsymbol{\alpha}' - \boldsymbol{\alpha}\|) \tag{8}$$

and the following alternative form for the conditional variance (see Appendix B for a derivation)

$$\mathrm{Var}_{p(\boldsymbol{\alpha})} [F(\boldsymbol{\alpha},\boldsymbol{\beta},\mathbf{x},\mathbf{e}) \mid \mathbf{x},\boldsymbol{\beta},\mathbf{e}] = \tfrac{1}{2}\mathbb{E}_{p(\boldsymbol{\alpha})p(\boldsymbol{\alpha}')} \left[ (F(\boldsymbol{\alpha},\boldsymbol{\beta},\mathbf{x},\mathbf{e}) - F(\boldsymbol{\alpha}',\boldsymbol{\beta},\mathbf{x},\mathbf{e}))^2 \mid \mathbf{x},\boldsymbol{\beta},\mathbf{e} \right] \tag{9}$$

Combining these two ideas, we have

$$V = \mathbb{E}_{p(\mathbf{x})p(\boldsymbol{\beta})p(\mathbf{e})} \left[ \tfrac{1}{2}\mathbb{E}_{p(\boldsymbol{\alpha})p(\boldsymbol{\alpha}')} \left[ (F(\boldsymbol{\alpha},\boldsymbol{\beta},\mathbf{x},\mathbf{e}) - F(\boldsymbol{\alpha}',\boldsymbol{\beta},\mathbf{x},\mathbf{e}))^2 \mid \mathbf{x},\boldsymbol{\beta},\mathbf{e} \right] \right] \tag{10}$$

$$\approx \mathbb{E}_{p(\mathbf{x})p(\boldsymbol{\beta})p(\mathbf{e})} \left[ \tfrac{1}{2}\mathbb{E}_{p(\boldsymbol{\alpha})p(\boldsymbol{\alpha}')} \left[ (\nabla_{\boldsymbol{\alpha}}F(\boldsymbol{\alpha},\boldsymbol{\beta},\mathbf{x},\mathbf{e}) \cdot (\boldsymbol{\alpha}' - \boldsymbol{\alpha}))^2 \mid \mathbf{x},\boldsymbol{\beta},\mathbf{e} \right] \right]. \tag{11}$$

Here we have an approximation of the conditional variance $V$ that uses gradient information. Including this as a regularizer within contrastive learning will encourage the encoder to reduce the magnitude of the conditional variance $V$, forcing the representation to change slowly as the transformation is varied and thus inducing approximate invariance to the transformations.

An unbiased estimator of equation 11 using a batch $\mathbf{x}_1, ..., \mathbf{x}_K$ is

$$\hat{V}_{\text{regularizer}} = \frac{1}{K} \sum_{i=1}^{K} \left( \frac{1}{2L} \sum_{j=1}^{L} \left[ \nabla_{\boldsymbol{\alpha}}F(\boldsymbol{\alpha}_i,\boldsymbol{\beta}_i,\mathbf{x}_i,\mathbf{e}_i) \cdot (\boldsymbol{\alpha}'_{ij} - \boldsymbol{\alpha}_i) \right]^2 \right) \tag{12}$$

where $\mathbf{x}_i,\boldsymbol{\alpha}_i,\boldsymbol{\beta}_i,\mathbf{e}_i, \sim p(\mathbf{x})p(\boldsymbol{\alpha})p(\boldsymbol{\beta})p(\mathbf{e})$, $\boldsymbol{\alpha}'_{ij} \sim p(\boldsymbol{\alpha})$. We can cheaply use a large number of samples for $\boldsymbol{\alpha}'$ without having to take any additional forward passes through the encoder: we only require $K$ evaluations of $F$. Our final loss function is

$$\mathcal{L}(\theta, \phi) = -\frac{1}{K} \sum_{i=1}^{K} s_\phi(\mathbf{z}_i, \mathbf{z}'_i) + \frac{1}{K} \sum_{i=1}^{K} \log \left( \sum_{j=1}^{K} \exp \left[ s_\phi(\mathbf{z}_i, \mathbf{z}'_j) \right] \right)$$

$$+ \frac{\lambda}{LK} \sum_{i=1}^{K} \sum_{j=1}^{L} \left[ \nabla_{\boldsymbol{\alpha}}F(\boldsymbol{\alpha}_i,\boldsymbol{\beta}_i,\mathbf{x}_i,\mathbf{e}_i) \cdot (\boldsymbol{\alpha}'_{ij} - \boldsymbol{\alpha}_i) \right]^2 \tag{13}$$

---

[2] We use the notation $a(x) = o(b(x))$ to mean $a(x)/b(x) \to 0$ as $x \to \infty$.

where $\lambda$ is a hyperparameter controlling the regularization strength. This loss does not require us to encode a larger number of differently transformed inputs. Instead, it uses the gradient at $(\mathbf{x}, \boldsymbol{\alpha}, \boldsymbol{\beta}, \mathbf{e})$ to control properties of the encoder in a neighbourhood of $\boldsymbol{\alpha}$. This can effectively reduce the representation gradient along the directions corresponding to many different transformations. This, in turn, creates an encoder that is approximately invariant to the transformations.

## 4 BETTER TEST TIME REPRESENTATIONS WITH FEATURE AVERAGING

At test time, standard practice (Hjelm et al., 2018; Kolesnikov et al., 2019) dictates that test representations be produced by applying the encoder to untransformed inputs (possibly using a central crop). It may be beneficial, however, to aggregate information from differently transformed versions of inputs to enforce invariance more directly, particularly when our previously introduced gradient regularization can only be applied to a subset of the transformation parameters. Furthermore, in real-world applications, it may not be possible to remove nuisance transformations at test time or, as in our Spirograph dataset, there may not be only one unique 'untransformed' version of $\mathbf{x}$.

To this end, we propose combining representations from different transformations using feature averaging. This approach, akin to ensembling, does not directly use one encoding from the network $\mathbf{f}_\theta$ as a representation for an input $\mathbf{x}$. Instead, we sample transformation parameters $\boldsymbol{\alpha}_1, ..., \boldsymbol{\alpha}_M \sim p(\boldsymbol{\alpha}), \boldsymbol{\beta}_1, ..., \boldsymbol{\beta}_M \sim p(\boldsymbol{\beta})$ independently, and average the encodings of these differently transformed versions of $\mathbf{x}$ to give a single feature averaged representation

$$\mathbf{z}^{(M)}(\mathbf{x}) = \frac{1}{M} \sum_{m=1}^{M} \mathbf{f}_\theta(\mathbf{t}(\mathbf{x}, \boldsymbol{\alpha}_m, \boldsymbol{\beta}_m)). \tag{14}$$

Using $\mathbf{z}^{(M)}$ aggregates information about $\mathbf{x}$ by averaging over a range of possible transformations, thereby directly encouraging invariance. Indeed, the resulting representation has lower conditional variance than the single-sample alternative, since

$$\mathrm{Var}_{p(\boldsymbol{\alpha}_{1:M})p(\boldsymbol{\beta}_{1:M})} \left[ \mathbf{e} \cdot \mathbf{z}^{(M)}(\mathbf{x}) \Big| \mathbf{x}, \mathbf{e} \right] = \frac{1}{M} \mathrm{Var}_{p(\boldsymbol{\alpha}_1)p(\boldsymbol{\beta}_1)} \left[ \mathbf{e} \cdot \mathbf{z}^{(1)}(\mathbf{x}) \Big| \mathbf{x}, \mathbf{e} \right]. \tag{15}$$

Further, unlike gradient regularization, this approach takes account of *all* transformations, including those which we cannot differentiate with respect to (e.g. left–right flip). It therefore forms a natural test time counterpart to our training methodology to promote invariance.

We do not recommend using feature averaged representations during training. During training, we need a training signal to recognize similar and dissimilar representations, and feature averaging will weaken this training signal. Furthermore, the computational cost of additional encoder passes is modest when used once at test time, but more significant when used at every training iteration.

As a test time tool though, feature averaging is powerful. In Theorem 1 below, we show that for certain downstream tasks the feature averaged representation will always perform better than the single-sample *transformed* alternative. The proof is presented in Appendix C.

**Theorem 1.** *Consider evaluation on a downstream task by fitting a linear classification model with softmax loss or a linear regression model with square error loss on with representations as features. For a fixed classifier or regressor and $M' \geq M$ we have*

$$\mathbb{E}_{p(\mathbf{x},y)p(\boldsymbol{\alpha}_{1:M'})p(\boldsymbol{\beta}_{1:M'})} \left[ \ell\left(\mathbf{z}^{(M')}, y\right) \right] \leq \mathbb{E}_{p(\mathbf{x},y)p(\boldsymbol{\alpha}_{1:M})p(\boldsymbol{\beta}_{1:M})} \left[ \ell\left(\mathbf{z}^{(M)}, y\right) \right]. \tag{16}$$

Empirically we find that, using the same encoder and the same linear classification model, feature averaging can outperform evaluation using *untransformed* inputs. That is, even when it is possible to remove the transformations at test time, it is beneficial to retain them and use feature averaging.

## 5 RELATED WORK

Contrastive learning (van den Oord et al., 2018; Hénaff et al., 2019) has progressively refined the role of transformations in learning representations, with Bachman et al. (2019) applying repeated data augmentation and Tian et al. (2019) using $Lab$ colour decomposition to define powerful self-supervised tasks. The range of transformations has progressively increased (Chen et al., 2020a;b),

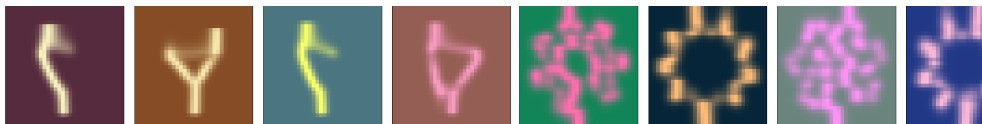

Figure 1: Samples from the spirograph dataset. Two sets of four images (left and right): each set shows different transformations applied to the same generative factors of interest.

whilst changing transformations can markedly improve performance (Chen et al., 2020c). Recent work has attempted to further understand and refine the role of transformations (Tian et al., 2020).

The idea of differentiating with respect to transformation parameters dates back to the tangent propagation algorithm (Simard et al., 1998; Rifai et al., 2011). Using the notation of this paper, tangent propagation penalizes the norm of the gradient of a neural network evaluated at $\boldsymbol{\alpha} = \mathbf{0}$, encouraging local transformation invariance near the original input. In our work, we target the conditional variance (Equation 6), leading to gradient evaluations across the $\boldsymbol{\alpha}$ parameter space with random $\boldsymbol{\alpha} \sim p(\boldsymbol{\alpha})$ and a regularizer that is not a gradient norm (Equation 12).

Our gradient regularization approach also connects to work on gradient regularization for Lipschitz constraints. A small Lipschitz constant has been shown to lead to better generalization (Sokolić et al., 2017) and improved adversarial robustness (Cisse et al., 2017; Tsuzuku et al., 2018; Barrett et al., 2021). Previous work focuses on constraining the mapping $\mathbf{x} \mapsto \mathbf{z}$ to have a small Lipschitz constant which is beneficial for *adversarial* robustness. In our work we focus on the influence of $\boldsymbol{\alpha}$ on $\mathbf{z}$, which gives rise to *transformation* robustness. Appendix D provides a more comprehensive discussion of related work.

## 6 EXPERIMENTS

### 6.1 DATASETS AND SET-UP

The methods proposed in this paper learn representations that discard some information, whilst retaining what is relevant. To more deeply explore this idea, we construct a dataset from a generative process controlled by both *generative factors of interest* and *nuisance transformations*. Representations should be able to recover the factors of interest, whilst being approximately invariant to transformation. To aid direct evaluation of this, we introduce a new dataset, which we refer to as the Spirograph dataset. Its samples are created using four generative factors and six nuisance transformation parameters. Figure 1 shows two sets of four samples with the generative factors fixed in each set. Every Spirograph sample is based on a hypotrochoid—one of a parametric family of curves that describe the path traced out by a point on one circle rolling around inside another. This generative process is fully differentiable in the parameters, meaning that our gradient regularization can be applied to every transformation. We define four downstream tasks for this dataset, each corresponding to the recovery of one of the four generative factors of interest using linear regression. The final dataset consists of 100k training and 20k test images of size $32 \times 32$. For full details of this dataset, see Appendix E.

As well as the Spirograph dataset, we apply our ideas to CIFAR-10 and CIFAR-100 (Krizhevsky et al., 2009). We base our contrastive learning set-up on SimCLR (Chen et al., 2020a). To use our gradient regularization, we adapt colour distortion (brightness, contrast, saturation and hue adjustment) as a fully differentiable transformation giving a four dimensional $\boldsymbol{\alpha}$; we also included random cropping and flipping but we did not apply a gradient regularization to these. We used ResNet50 (He et al., 2016) encoders for CIFAR and ResNet18 for Spirograph, and regularization parameters $\lambda = 0.1$ for CIFAR and $\lambda = 0.01$ for Spirograph. For comprehensive details of our set-up and additional plots, see Appendix F. For an open source implementation of our methods, see `https://github.com/ae-foster/invclr`.

### 6.2 GRADIENT REGULARIZATION LEADS TO STRONGLY INVARIANT REPRESENTATIONS

We first show that our gradient penalty successfuly learns representations that are more invariant to transformation than standard contrastive learning. First, we estimate the conditional variance

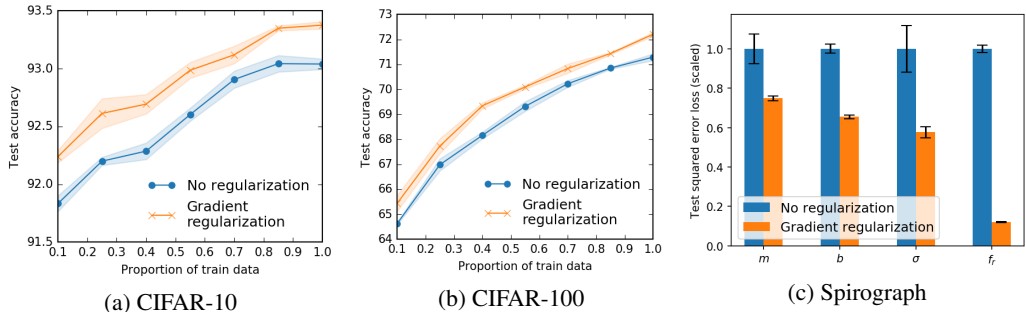

|                |                |                |
|:--------------:|:--------------:|:--------------:|
| (a) CIFAR-10   | (b) CIFAR-100  | (c) Spirograph |

Figure 3: Downstream task performance of gradient regularized representations. (a)(b) Top-1 test accuracy for various levels of semi-supervision (higher better). (c) Test loss on four downstream regression tasks on Spirograph that recover the generative factors of interest (lower better). The loss is rescaled for legibility, see Table 9 for raw values. Error bars are $\pm 1$ standard error from 3 runs.

of the representation that was used as the starting point for motivating our approach, i.e. Equation 6, using the slower, but more exact, nested Monte Carlo estimator of Equation 7 to evaluate this. In Figure 2 we see that the gradient penalty strikingly reduces the conditional variance on CIFAR-10 compared to standard contrastive learning.

As an additional measure of representation invariance, we fit a linear regression model that predicts $\alpha$ from $z$, for which higher loss indicates a greater degree of invariance. We also compute a reference loss: the loss that would be obtained when predicting $\alpha$ using only a constant. In Table 1, we see that unlike standard contrastive learning, after training with gradient regularization the linear regression model cannot predict $\alpha$ from $z$ any better than using a constant prediction. The loss is actually higher than the reference value because the former is obtained by training a regressor for a finite number of steps, whilst the latter is a theoretical optimum value. Similar results for other datasets are in Appendix F.

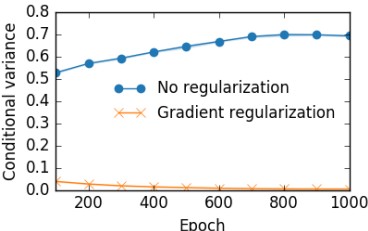

Figure 2: Conditional variance for CIFAR-10 as per Equation 6. Error bars represent $\pm 1$ standard error from 3 runs.

Table 1: Test loss when linear regression is used to predict $\alpha$ from $z$ on CIFAR-10. The reference value is $\mathrm{Mean}_i \mathrm{Var}(\alpha_i)$. We present the mean and $\pm 1$ s.e. from 3 runs.

|                   | Test loss           |
|-------------------|---------------------|
| No regularization | $0.0353 \pm 0.0002$ |
| Regularization    | $0.0415 \pm 0.00006$ |
| Reference value   | $0.0408$            |

### 6.3 GRADIENT REGULARIZATION FOR DOWNSTREAM TASKS AND TEST TIME DATASET SHIFT

We now show that these more invariant representations perform better on downstream tasks. For CIFAR, we produce representations for each element of the training and test set (by applying the encoder $\mathbf{f}_\theta$ to untransformed inputs). We then fit a linear classifier on the training set, using different fractions of the class labels. This allows us to assess our representations at different levels of supervision. We use the entire test set to evaluate each of these classifiers. In Figures 3(a) and (b), we see that the test accuracy improves across the board with gradient regularization.

For Spirograph, we take a similar approach to evaluation: we create representations for the training and test sets and fit linear regression models with representations as features for each of the four downstream tasks. In Figure 3(c), we see the test loss on each task with the baseline scaled to 1. Here we see huge improvements across all tasks, presumably due to the ability to apply gradient regularization to all transformations (unlike for CIFAR).

We further study the effect of transformation at test time, showing that gradient penalized representations can be more robust to shifts in the transformation distribution. For CIFAR-10, we apply colour distortion transformations at test time with different levels of variance. By focusing on colour distortion at test time, we isolate the transformations that the gradient regularization targeted. In Figure 4(a) we see that when the test time distribution is shifted to have higher variance than the training regime, our gradient penalized representations perform better than using contrastive learn-

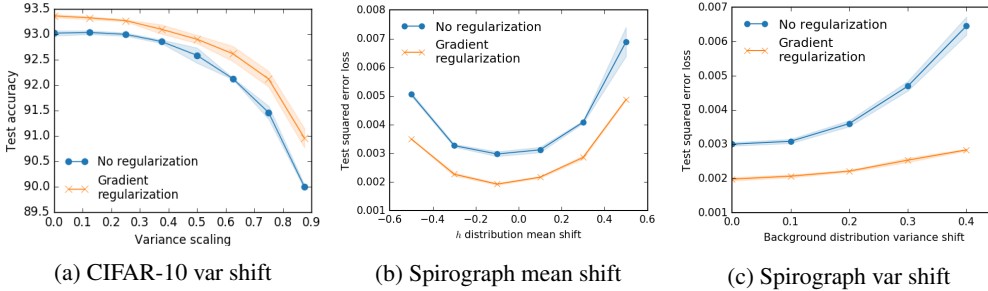

| (a) CIFAR-10 var shift | (b) Spirograph mean shift | (c) Spirograph var shift |

Figure 4: Assessing representation robustness to test time distribution shift. (a) Changing the variance of colour distortions; 0 is no transformation and 0.5 is the training regime. (b) Mean shifting of the distribution of the transformation parameter $h$. (c) Variance shifting of the background colour distribution. In (b)(c), 0 shift indicates the training regime. Error bars are $\pm 1$ s.e. from 3 runs.

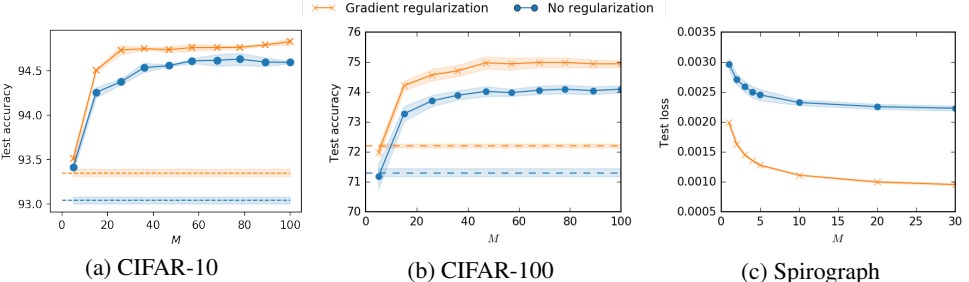

| (a) CIFAR-10 | (b) CIFAR-100 | (c) Spirograph |

Figure 5: The impact of feature averaging on CIFAR-10, CIFAR-100 and Spirograph. (a)(b) Test accuracy for CIFAR-10 and CIFAR-100 respectively for various values of $M$. Dashed lines represent evaluation with untransformed inputs. (c) Test mean square error averaged over all four tasks for Spirograph (untransformed inputs not valid here). Error bars are $\pm 1$ s.e. from 3 runs.

ing alone. For Spirograph, we investigate changing both the mean of the transformation distribution, moving the entire test distribution away from the training regime, and increasing the variance of transformations to add noise. Results are shown in Figure 4(b) and (c). In 4(c) in particular, we see that gradient regularized representations are robust to a greater level of distortion at test time.

## 6.4 FEATURE AVERAGING FURTHER IMPROVES PERFORMANCE

We now assess the impact of feature averaging on test time performance. For CIFAR, we apply feature averaging using *all* transformations, including random crops etc., and compare with the standard protocol of using untransformed inputs to form the test representations. Figures 5(a) and (b) show that feature averaging leads to significant improvements. This adds to the result of Theorem 1, which implies that test loss decreases as $M$ is increased. In Figure 5(c), we see that feature averaging has an equally beneficial impact on Spirograph. It is interesting to note that in both cases there is still significant residual benefit from gradient regularization, even with a large value of $M$.

## 6.5 OUR METHODS COMPARE FAVOURABLY WITH OTHER PUBLISHED BASELINES

Our primary aim was to show that both gradient regularization and feature averaging lead to improvements compared to baselines that are in other respects identical. Our methods are applicable to almost any base contrastive learning approach, and we would expect them to deliver improvements across this range of different base methods. In Table 2, we present published baselines on

Table 2: Comparative best test accuracy of various self-supervized representation learning techniques, evaluated using linear classification.

| Method | CIFAR-10 acc. | CIFAR-100 acc. |
|---|---|---|
| AMDIM small (Bachman et al., 2019) | 89.5% | 68.1% |
| AMDIM large (Bachman et al., 2019) | 91.2% | 70.2% |
| SimCLR (Chen et al., 2020a) | 94.0% | - |
| **Ours (SimCLR base)** | **94.9**% | **75.1**% |

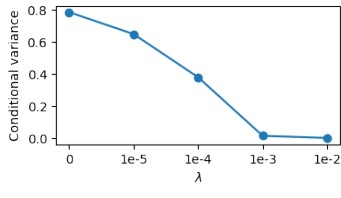

(a) Conditional variance

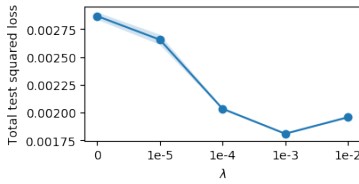

(b) Downstream task performance

Figure 6: The impact of the regularization hyperparameter $\lambda$ on representation learning with the Spirograph dataset. (a) Conditional variance of Equation 6. (b) The total mean square error on all four downstream tasks. Error bars are $\pm 1$ s.e. from 3 runs. Smaller is better in both cases giving an optimum around $\lambda = 10^{-3}$, but with stable performance as $\lambda$ is increased above this.

Table 3: Results for representation learning on CIFAR-100 with MoCo v2 as the base contrastive learning method, with gradient regularization in isolation and in combination with feature averaging. We trained two different encoder architectures. We present test accuracy from linear classification evaluation. Feature averaging uses $M = 40$. Errors are $\pm 1$ s.e. from multiple runs.

| Method | ResNet18 acc. | ResNet50 acc. |
|---|---|---|
| MoCo v2 | $52.3\% \pm 0.3$ | $57.9\% \pm 0.1$ |
| MoCo v2 with gradient penalty | $54.1\% \pm 0.1$ | $58.9\% \pm 0.2$ |
| **MoCo v2 with gradient penalty and feature averaging** | **$60.6\% \pm 0.1$** | **$64.4\% \pm 0.2$** |

CIFAR datasets, along with the results that we obtain using our gradient regularization and feature averaging with SimCLR as a base method. This is the default base method that we recommend, and that was used in our previous experiments. Interestingly, the best ResNet50 encoder from our experiments achieves an accuracy of 94.9% on CIFAR-10, which outperforms the next best published result from the contrastive learning literature by almost 1%, and 75.1% on CIFAR-100, an almost 5% improvement over a significantly larger encoder architecture. As such, we see our results actually provide performance that is state-of-the-art for contrastive learning on these benchmarks. In fact, our performance increases almost entirely close the gap to the state-of-the-art performance for fully supervized training with the same architecture on CIFAR-10 (95.1%, Chen et al. (2020a)).

To demonstrate that our ideas generalize to other contrastive learning base methods, we apply our ideas to MoCo v2 (Chen et al., 2020c). Table 3 shows that, whilst MoCo v2 itself does not perform as well as SimCLR on CIFAR-100, the addition of gradient regularization and feature averaging still leads to significant improvements in its performance. Table 3 further illustrates that both gradient regularization and feature averaging contribute to the performance improvements offered by our approach and that our techniques generalize across diffrent encoder architectures.

## 6.6 HYPERPARAMETER SENSITIVITY

As a further ablation study, we investigated the sensitivity of our method to changes in the gradient regularization hyperparameter $\lambda$ (as defined in Equation 13). In Figure 6(a) we see that, as expected, the conditional variance of representations decreases as $\lambda$ is increased. The downstream task performance Figure 6(b) similarly improves as we increase $\lambda$, reaching an optimum around $\lambda = 10^{-3}$, before beginning to increase due to over-regularization. We see that *a wide range of values* of $\lambda$ deliver good performance and the method is not overly sensitive to careful tuning of $\lambda$.

## 7 CONCLUSION

Viewing contrastive representation learning through the lens of representation invariance to transformation, we derived a gradient regularizer that controls how quickly representations can change with transformation, and proposed feature averaging at test time to pull in information from multiple transformations. These approaches led to representations that performed better on downstream tasks. Therefore, our work provides evidence that invariance is highly relevant to the success of contrastive learning methods, and that there is scope to further improve upon these methods by using invariance as a guiding principle.

ACKNOWLEDGMENTS

AF gratefully acknowledges funding from EPSRC grant no. EP/N509711/1. AF would also like to thank Benjamin Bloem-Reddy for helpful discussions about theoretical aspects of this work.

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

## A  MUTUAL INFORMATION

In Section 2, we saw that the InfoNCE objective equation 3 fulfills the need to make $p(\mathbf{z}|\mathbf{x})$ tightly focused on a single point whilst simultaneously requiring $p(\mathbf{z})$ to be well spread out over representation space. In this appendix, we show that the same general principle of making $p(\mathbf{z}|\mathbf{x})$ tightly focused on a single point whilst simultaneously requiring $p(\mathbf{z})$ to be well spread out over representation space connects to mutual information maximization.

To establish the connection to mutual information, we take the differential entropy as our measure of 'spread'. Recall the differential entropy of a random variable $\mathbf{w}$ is

$$H[p(\mathbf{w})] := \mathbb{E}_{p(\mathbf{w})}[-\log p(\mathbf{w})]. \tag{17}$$

We then translate our intuition to make $p(\mathbf{z}|\mathbf{x})$ tightly focused on a single point whilst simultaneously requiring $p(\mathbf{z})$ to be well spread out over representation space into requiring $\mathbb{E}_{p(\mathbf{x})}[H[p(\mathbf{z}|\mathbf{x})]]$ to be minimized whilst $H[p(\mathbf{z})]$ should be simultaneously maximized. This suggests the following loss function

$$\mathcal{L}_{\text{entropy}} = \mathbb{E}_{p(\mathbf{x})}[H[p(\mathbf{z}|\mathbf{x})]] - H[p(\mathbf{z})] = -I(\mathbf{x}; \mathbf{z}) \tag{18}$$

which is the (negative) mutual information between $\mathbf{x}$ and $\mathbf{z}$. Note that in this formulation, it is the distribution $p(\mathbf{z}|\mathbf{x})$ as much as the InfoMax principle which determines how this loss will behave. Finally, there is a clear connection between the InfoNCE loss and mutual information, specifically the InfoNCE loss is, in expectation and up to an additive constant, a lower bound on $I(\mathbf{x}; \mathbf{z})$ van den Oord et al. (2018); Poole et al. (2019).

## B  METHOD

### B.1  AN ALTERNATIVE VARIANCE FORMULA

We present a derivation of our alternative formula for the variance (dropping the conditioning from the notation for conciseness)

$\frac{1}{2}\mathbb{E}_{p(\boldsymbol{\alpha})p(\boldsymbol{\alpha}')}\left[(F(\boldsymbol{\alpha}, \boldsymbol{\beta}, \mathbf{x}, \mathbf{e}) - F(\boldsymbol{\alpha}', \boldsymbol{\beta}, \mathbf{x}, \mathbf{e}))^2\right]$

$= \frac{1}{2}\mathbb{E}_{p(\boldsymbol{\alpha})p(\boldsymbol{\alpha}')}\left[(F(\boldsymbol{\alpha}, \boldsymbol{\beta}, \mathbf{x}, \mathbf{e}) - \mathbb{E}_{\boldsymbol{\alpha}}[F(\boldsymbol{\alpha}, \boldsymbol{\beta}, \mathbf{x}, \mathbf{e})] + \mathbb{E}_{\boldsymbol{\alpha}}[F(\boldsymbol{\alpha}, \boldsymbol{\beta}, \mathbf{x}, \mathbf{e})] - F(\boldsymbol{\alpha}', \boldsymbol{\beta}, \mathbf{x}, \mathbf{e}))^2\right]$

$= \frac{1}{2}\mathbb{E}_{p(\boldsymbol{\alpha})p(\boldsymbol{\alpha}')}\left[(F(\boldsymbol{\alpha}, \boldsymbol{\beta}, \mathbf{x}, \mathbf{e}) - \mathbb{E}_{\boldsymbol{\alpha}}[F(\boldsymbol{\alpha}, \boldsymbol{\beta}, \mathbf{x}, \mathbf{e})])^2 + (\mathbb{E}_{\boldsymbol{\alpha}}[F(\boldsymbol{\alpha}, \boldsymbol{\beta}, \mathbf{x}, \mathbf{e})] - F(\boldsymbol{\alpha}', \boldsymbol{\beta}, \mathbf{x}, \mathbf{e}))^2\right]$

$\quad + \mathbb{E}_{p(\boldsymbol{\alpha})p(\boldsymbol{\alpha}')}\left[(F(\boldsymbol{\alpha}, \boldsymbol{\beta}, \mathbf{x}, \mathbf{e}) - \mathbb{E}_{\boldsymbol{\alpha}}[F(\boldsymbol{\alpha}, \boldsymbol{\beta}, \mathbf{x}, \mathbf{e})])(\mathbb{E}_{\boldsymbol{\alpha}}[F(\boldsymbol{\alpha}, \boldsymbol{\beta}, \mathbf{x}, \mathbf{e})] - F(\boldsymbol{\alpha}', \boldsymbol{\beta}, \mathbf{x}, \mathbf{e}))\right]$

$= \frac{1}{2}\mathbb{E}_{p(\boldsymbol{\alpha})p(\boldsymbol{\alpha}')}\left[(F(\boldsymbol{\alpha}, \boldsymbol{\beta}, \mathbf{x}, \mathbf{e}) - \mathbb{E}_{\boldsymbol{\alpha}}[F(\boldsymbol{\alpha}, \boldsymbol{\beta}, \mathbf{x}, \mathbf{e})])^2 + (F(\boldsymbol{\alpha}', \boldsymbol{\beta}, \mathbf{x}, \mathbf{e}) - \mathbb{E}_{\boldsymbol{\alpha}}[F(\boldsymbol{\alpha}, \boldsymbol{\beta}, \mathbf{x}, \mathbf{e})])^2\right]$

$= \text{Var}_{p(\boldsymbol{\alpha})}[F(\boldsymbol{\alpha}, \boldsymbol{\beta}, \mathbf{x}, \mathbf{e})].$

### B.2  MOTIVATING THE RADEMACHER DISTRIBUTION

We are interested in the conditional variance of $\mathbf{z}$ with respect to $\boldsymbol{\alpha}$, but as $\mathbf{z}$ is a vector valued random variable we properly need to consider the conditional covariance matrix $\Sigma = \text{Cov}_{\boldsymbol{\alpha}}(\mathbf{z}|\mathbf{x}, \boldsymbol{\beta})$. We henceforth consider $\mathbf{x}, \boldsymbol{\beta}$ to be fixed. To reduce conditional variance in all directions, it makes sense to reduce the trace $\text{Tr}\,\Sigma$. Due to computational limitations, we cannot directly estimate this trace at each iteration, instead we must estimate $\text{Var}(\mathbf{e} \cdot \mathbf{z}) = \mathbf{e}^\top \Sigma \mathbf{e}$. However, by carefully selecting the distribution for $\mathbf{e}$ we can effectively target the trace of the covariance matrix by taking the expectation over $\mathbf{e}$. Specifically, suppose that the components of $\mathbf{e}$ are independent Rademacher random variables ($\pm 1$ with equal probability). Then

$$\mathbb{E}_{p(\mathbf{e})}\left[\mathbf{e}^\top \Sigma \mathbf{e}\right] = \mathbb{E}_{p(\mathbf{e})}\left[\sum_{ij} e_i \Sigma_{ij} e_j\right] = \sum_{ij} \Sigma_{ij} \mathbb{E}_{p(\mathbf{e})}[e_i e_j] = \sum_{ij} \Sigma_{ij} \delta_{ij} = \text{Tr}\,\Sigma. \tag{19}$$

## C  THEORY

We present the proof of Theorem 1 which is restated for convenience.

**Theorem 1.** *Consider evaluation on a downstream task by fitting a linear classification model with softmax loss or a linear regression model with square error loss on with representations as features. For a fixed classifier or regressor and $M' \geq M$ we have*

$$\mathbb{E}_{p(\mathbf{x},y)p(\boldsymbol{\alpha}_{1:M'})p(\boldsymbol{\beta}_{1:M'})} \left[ \ell\left(\mathbf{z}^{(M')}, y\right) \right] \leq \mathbb{E}_{p(\mathbf{x},y)p(\boldsymbol{\alpha}_{1:M})p(\boldsymbol{\beta}_{1:M})} \left[ \ell\left(\mathbf{z}^{(M)}, y\right) \right]. \tag{16}$$

*Proof.* We have the softmax loss

$$\ell(\mathbf{z}, y) = -\boldsymbol{w}_y^\top \mathbf{z} + \log\left( \sum_j \exp\left(\boldsymbol{w}_j^\top \mathbf{z}\right) \right) \tag{20}$$

or the square error loss

$$\ell(\mathbf{z}, y) = \left| y - \boldsymbol{w}^\top \mathbf{z} \right|^2. \tag{21}$$

We first show that both loss functions considered are convex in the argument $\mathbf{z}$. To show this, we fix $0 \leq p = 1 - q \leq 1$. For softmax loss, we have

$$\ell\left(p\mathbf{z}_1 + q\mathbf{z}_2, y\right) \tag{22}$$

$$= -\mathbf{w}_y^\top(p\mathbf{z}_1 + q\mathbf{z}_2) + \log\left( \sum_j \exp\left(\mathbf{w}_j^\top(p\mathbf{z}_1 + q\mathbf{z}_2)\right) \right) \tag{23}$$

$$= -p\mathbf{w}_y^\top \mathbf{z}_1 - q\mathbf{w}_y^\top \mathbf{z}_2 + \log\left( \sum_j \exp\left(\mathbf{w}_j^\top \mathbf{z}_1\right)^p \exp\left(\mathbf{w}_j^\top \mathbf{z}_2\right)^q \right) \tag{24}$$

$$\leq -p\mathbf{w}_y^\top \mathbf{z}_1 - q\mathbf{w}_y^\top \mathbf{z}_2 + \log\left( \left(\sum_j \exp\left(\mathbf{w}_j^\top \mathbf{z}_1\right)\right)^p \left(\sum_j \exp\left(\mathbf{w}_j^\top \mathbf{z}_2\right)\right)^q \right) \tag{25}$$

by Hölder's Inequality

$$= -p\mathbf{w}_y^\top \mathbf{z}_1 - q\mathbf{w}_y^\top \mathbf{z}_2 + p\log\left( \sum_j \exp\left(\mathbf{w}_j^\top \mathbf{z}_1\right) \right) + q\log\left( \sum_j \exp\left(\mathbf{w}_j^\top \mathbf{z}_2\right) \right) \tag{26}$$

$$= p\ell\left(\mathbf{z}_1, y\right) + q\ell\left(\mathbf{z}_2, y\right) \tag{27}$$

and for square error loss we have

$$\ell\left(p\mathbf{z}_1 + q\mathbf{z}_2, y\right) = |y - \boldsymbol{w}^\top(p\mathbf{z}_1 + q\mathbf{z}_2)|^2 \tag{28}$$

$$= |p(y - \boldsymbol{w}^\top \mathbf{z}_1) + q(y - \boldsymbol{w}^\top \mathbf{z}_2)|^2 \tag{29}$$

$$= p|y - \boldsymbol{w}^\top \mathbf{z}_1|^2 + q|y - \boldsymbol{w}^\top \mathbf{z}_2|^2 + (p^2 - p)\left|\boldsymbol{w}^\top \mathbf{z}_1 - \boldsymbol{w}^\top \mathbf{z}_2\right|^2 \tag{30}$$

$$\leq p|y - \boldsymbol{w}^\top \mathbf{z}_1|^2 + q|y - \boldsymbol{w}^\top \mathbf{z}_2|^2 \tag{31}$$

$$= p\ell\left(\mathbf{z}_1, y\right) + q\ell\left(\mathbf{z}_2, y\right) \tag{32}$$

For the inequality in the Theorem, we consider drawing $M' \geq M$ samples, and randomly choosing an $M$-subset. Let $S$ represent this subset and let $\mathbf{z}_S^{(M)}$ represent the feature averaged representation that uses the subset $S$. We have

$$\mathbb{E}_{p(\mathbf{x},y)p(\boldsymbol{\alpha}_{1:M})p(\boldsymbol{\beta}_{1:M})} \left[ \ell\left(\mathbf{z}^{(M)}, y\right) \right] = \mathbb{E}_{p(\mathbf{x},y)p(\mathbf{t}_{1:M'})p(\boldsymbol{\alpha}_{1:M'})p(\boldsymbol{\beta}_{1:M'})p(S)} \left[ \ell\left(\mathbf{z}_S^{(M)}, y\right) \right] \tag{33}$$

$$\geq \mathbb{E}_{p(\mathbf{x},y)p(\boldsymbol{\alpha}_{1:M'})p(\boldsymbol{\beta}_{1:M'})} \left[ \ell\left(\mathbb{E}_{p(S)}\left[\mathbf{z}_S^{(M)}\right], y\right) \right] \tag{34}$$

$$= \mathbb{E}_{p(\mathbf{x},y)p(\boldsymbol{\alpha}_{1:M'})p(\boldsymbol{\beta}_{1:M'})} \left[ \ell\left(\mathbf{z}^{(M')}, y\right) \right] \tag{35}$$

where the inequality at equation 34 is by Jensen's Inequality. This completes the proof. ☐

We provide an informal discussion of other theoretical results that relate to our work. Lyle et al. (2020) explored PAC-Bayesian approaches to analyzing the role of group invariance in generalization of supervized neural network models. The central bound based on Catoni (2007) is given in

Theorem 1 of Lyle et al. (2020) and depends on the empirical risk $\hat{R}_\ell(Q, \mathcal{D}_n)$ and the KL term $KL(Q\|P)$ which represents the PAC-Bayesian KL divergence between distributions on hypothesis space. Theorem 7 of Lyle et al. (2020) shows $KL(Q^\circ\|P^\circ) \leq KL(Q\|P)$, where $Q^\circ$ and $P^\circ$ are formed by symmetrization such as feature averaging over the group of transformations. In our context, although the transformations do not form a group, we could still consider a symmetrization operation with feature averaging. If the symmetrization does not affect the empirical risk, then Theorem 9 of Lyle et al. (2020) would apply to our setting and we would be able to obtain a tighter generalization bound for our suggested approach of feature averaging.

## D  RELATED WORK

### D.1  THE ROLE OF TRANSFORMATIONS IN CONTRASTIVE LEARNING

Recent work on contrastive learning, initiated by the development of Contrastive Predictive Coding (van den Oord et al., 2018; Hénaff et al., 2019), has progressively moved the transformations to a more central position in understanding and improving these approaches. In Bachman et al. (2019), multiple views of a context are extracted, on images this utilizes repeated data augmentation such as random resized crop, random colour jitter, and random conversion to grayscale, and the model is trained to maximize information between these views using an InfoNCE style objective. Other approaches are possible, for instance Tian et al. (2019) obtained multiple views of images using $Lab$ colour decomposition. In SimCLR (Chen et al., 2020a;b), the approach of applying multiple data augmentations (including flip and blur, as well as crops, colour jitter and random grayscale) and using an InfoNCE objective was simplified and streamlined, and the central role of the augmentations was emphasized. By changing the set of transformation operations used, Chen et al. (2020c) were able to improve their contrastive learning approach and achieve excellent performance on downstream detection and segmentation tasks. Tian et al. (2020) studied what the best range of transformations for contrastive learning is. The authors found that there is a 'sweet spot' in the strength of transformations applied in contrastive learning, with transformations that are too strong or weak being less favourable. Winkens et al. (2020) showed that contrastive methods can be successfully applied to out-of-distribution detection. We note that for tasks such as out-of-distribution detection, transformation *covariance* may be a more relevant property than invariance.

### D.2  GRADIENT REGULARIZATION TO ENFORCE LIPSCHITZ CONSTRAINTS

Constraining a neural network to be Lipschitz continuous bounds how quickly its output can change as the input changes. In supervised learning, a small Lipschitz constant has been shown to lead to better generalization (Sokolić et al., 2017) and improved adversarial robustness (Cisse et al., 2017; Tsuzuku et al., 2018). One practical method for constraining the Lipschitz constant is gradient regularization (Drucker & Le Cun, 1992; Gulrajani et al., 2017). Lipschitz constraints have also been applied in a self-supervised context: in Ozair et al. (2019), the authors used a Wasserstein dependency measure in a contrastive learning setting by using gradient penalization to ensure that the function $\mathbf{x}, \mathbf{x}' \mapsto s_\phi(\mathbf{f}_\theta(\mathbf{x}), \mathbf{f}_\theta(\mathbf{x}'))$ is 1-Lipschitz. Our work uses a gradient regularizer to control how quickly representations can change, but unlike existing work we focus on how representations change with $\boldsymbol{\alpha}$ as $\mathbf{x}$ is fixed, instead of how they change with $\mathbf{x}$.

### D.3  GROUP INVARIANT NEURAL NETWORKS

A large body of recent work has focused on designing neural network architectures that are perfectly invariant, or equivariant, to a set of transformations $\mathcal{T}$ in the case when $\mathcal{T}$ forms a group. Cohen & Welling (2016) showed how convolutional neural networks can be generalized to have equivariance to arbitrary group transformations applied to their inputs. This can apply, for instance, to rotation groups on the sphere (Cohen et al., 2018), rotation and translation groups on point clouds (Thomas et al., 2018), and permutation groups on sets (Zaheer et al., 2017). Transformations that form a group cannot remove information from the input (because they must be invertible) and can be composed in any order. This means that the more general transformations considered in our work cannot form a group—they cannot be composed (repeated decreasing of brightness to zero is not allowed) nor inverted (crops are not invertible). We have therefore considered methods that improve invariance under much more general transformations.

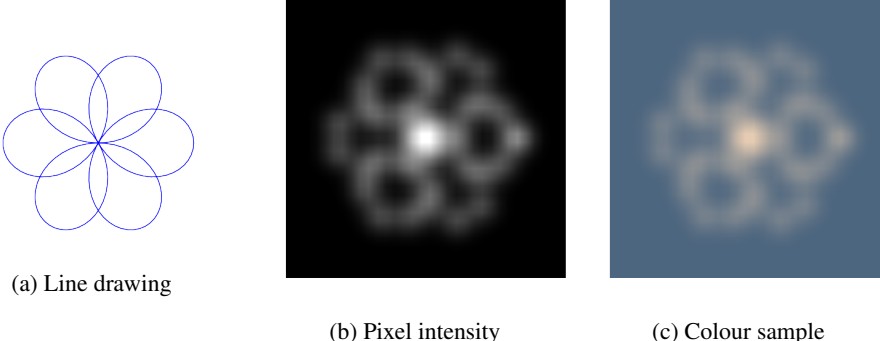

(a) Line drawing

(b) Pixel intensity  (c) Colour sample

Figure 7: A sample from the Spirograph dataset with $m = 4, b = 0.4, h = 2, \sigma = 1, (f_r \quad f_g \quad f_b) = (0.9 \quad 0.8 \quad 0.7), (b_r \quad b_g \quad b_b) = (0.3 \quad 0.4 \quad 0.5)$.

### D.4 FEATURE AVERAGING AND POOLING

The concepts of sum-, max- and mean-pooling have a rich history in deep learning (Krizhevsky et al., 2012; Graham, 2014). For example, pooling can be used to down-scale representations in convolutional neural networks (CNNs) as part of a single forward pass through the network with a single input. In our work, however, we apply feature averaging, or mean-pooling, using multiple, differently transformed versions of the same input. This is more similar to Chatfield et al. (2014), who considered pooling or stacking augmented inputs as part of a CNN, and Yoo et al. (2015) who proposed a multi-scale pyramid pooling approach. Unlike these works, we apply pooling in an unsupervised contrastive representation learning context. Our feature averaging occurs on the final representations, rather than in a pyramid, and not on intermediate layers of the network. We also use the transformation distribution that is used to define the self-supervised task itself. Other work has explored theoretical aspects of feature averaging (Chen et al., 2019; Lyle et al., 2020) in the supervised learning setting, showing conditions on the invariance properties of the underlying data distribution that can be exploited to obtain improved generalization using feature averaging. For a detailed discussion of Lyle et al. (2020) and its connections with our own work, see Section C.

## E SPIROGRAPH DATASET

We propose a new dataset that allows the separation of *generative factors of interest* from *nuisance transformation factors* and that is formed from a fully differentiable generative process. A standalone implementation of this dataset can be found at `https://github.com/rattaoup/spirograph`. Our dataset is inspired by the beautiful spirograph patterns some of us drew as children, which are mathematically hypotrochoids given by the following equations

$$x = (m - h)\cos(t) + h\cos\left(\frac{(m-h)t}{b}\right) \tag{36}$$

$$y = (m - h)\sin(t) - h\sin\left(\frac{(m-h)t}{b}\right) \tag{37}$$

Figure 7(a) shows an example. To create an image dataset from such curves, we choose 40 equally spaced points $t_i$ with $t_1 = 0$ and $t_{40} = 2\pi$, giving a sequence of points $(x_1, y_1), ..., (x_{40}, y_{40})$ on the chosen hypotrochoid. For smoothing parameter $\sigma$, the pixel intensity at a point $(u, v)$ is given by

$$i(u, v) = \frac{1}{40}\sum_{i=1}^{40}\exp\left(\frac{-(u - x_i)^2 - (v - y_i)^2}{\sigma}\right). \tag{38}$$

For a grid of pixel, the intensity values are normalized so that the maximum intensity is equal to 1. Figure 7(b) shows the pixel intensity with $\sigma = 0.5$. Finally, for a foreground colour with RGB values $(f_r, f_g, f_b)$ and background colour $(b_r, b_g, b_b)$, the final RGB values at a point $(u, v)$ is

$$\boldsymbol{c}(u, v) = i(u, v)\begin{pmatrix} f_r \\ f_g \\ f_b \end{pmatrix} + (1 - i(u, v))\begin{pmatrix} b_r \\ b_g \\ b_b \end{pmatrix} \tag{39}$$

The final coloured sample image is shown in Figure 7(c).

The Spirograph sample is fully specified by the parameters $m, b, h, \sigma, f_r, f_g, f_b, b_r, b_g, b_b$. In our experiments, we treat $m, b, \sigma, f_r$ as parameters of interest. We treat $h$ and the remaining colour parameters as nuisance parameters. That is, we take $\mathbf{x} = (m, b, \sigma, f_r)$ and $\boldsymbol{\alpha} = (h, f_g, f_b, b_r, b_g, b_b)$ and the transformation $\mathbf{t}(\mathbf{x}, \boldsymbol{\alpha})$ is the full generative process described above. There are no additional parameters $\boldsymbol{\beta}$ for this dataset. Figure 1 shows two sets of four samples from the Spirograph dataset, in each set the generative factors of interest are fixed and the nuisance parameters are varied. In general for the Spirograph dataset, the distinction between generative factors of interest and nuisance parameters can be changed to attempt to learn different aspects of the data. The transformation $\mathbf{t}$ is fully differentiable, meaning that we can apply gradient penalization to all the nuisance parameters of the generative process. In our experiments, we took the following distributions to sample random values of the parameters: $m \sim U(2, 5), b \sim U(0.1, 1.1), h \sim U(0.5, 2.5), \sigma \sim U(0.25, 1), f_r, f_g, f_b \sim U(0.4, 1), b_r, b_g, b_b \sim U(0, 0.6)$. We synthesized 100,000 training images and 20,000 test images with dimension $32 \times 32$.

## F    Experiment details

Our experiments were implemented in PyTorch (Paszke et al., 2019) and ran on 8 Nvidia GeForce GTX 1080Ti GPUs. See `https://github.com/ae-foster/invclr` for an implementation of our approaches.

### F.1    Differentiable colour distortion

We want to improve the representations learned from contrastive methods by explicitly encouraging stronger invariance to the set of transformations. Our method is to restrict gradients of the representations with respect to certain transformations. Ensuring that the transformations are practically differentiable within PyTorch (Paszke et al., 2019) required a thorough study of the transformations. The subset of transformations we apply gradient regularization to includes colour distortions which are conventionally treated as a part of data preprocessing. Rewriting this as a differentiable module within the computational graph allows us to practically compute the gradient regularizer of equation 11. We will consider adjusting brightness, contrast, saturation, hue of an image. In fact, most of these transformations are simply linear transformations of the original image. First, the brightness adjustment is simply defined as

$$\mathbf{x}_{\text{brt}} = \mathbf{x}\alpha_{\text{brt}} \tag{40}$$

when $\alpha_{\text{brt}}$ is a scale factor. If we write $\mathbf{x} = \mathbf{r}, \mathbf{g}, \mathbf{b}$, for the three colour channels of $\mathbf{x}$, then greyscale conversion of $\mathbf{x}$ is given by

$$\mathbf{x}_{\text{gs}} = 0.299\mathbf{r} + 0.587\mathbf{g} + 0.114\mathbf{b}. \tag{41}$$

Adjusting the saturation of $\mathbf{x}$ is a linear combination of $\mathbf{x}$ and $\mathbf{x}_{\text{gs}}$, the greyscale version of $\mathbf{x}$

$$\mathbf{x}_{\text{sat}} = \mathbf{x}\alpha_{\text{sat}} + \mathbf{x}_{\text{gs}}(1 - \alpha_{\text{sat}}) \tag{42}$$

when $\alpha_{\text{sat}}$ is a scale factor. Adjusting the contrast of $\mathbf{x}$ is a linear combination of $\mathbf{x}$ and mean($\mathbf{x}_{\text{gs}}$), which the mean over all spatial dimensions of $\mathbf{x}_{\text{gs}}$. With a scaling parameter $\alpha_{\text{con}}$ we have

$$\mathbf{x}_{\text{con}} = \mathbf{x}\alpha_{\text{con}} + \text{mean}(\mathbf{x}_{\text{gs}})(1 - \alpha_{\text{con}}). \tag{43}$$

We utilize a linear approximation for hue adjustment. We perform hue adjustment by converting to the YIQ colour space, and then applying rotation on the IQ components. The transformation between RGB and YIQ colour space is given by the following linear transformation

$$\begin{pmatrix} Y \\ I \\ Q \end{pmatrix} = \begin{pmatrix} 0.299 & 0.587 & 0.114 \\ 0.5959 & -0.2746 & -0.3213 \\ 0.2115 & -0.5227 & 0.3112 \end{pmatrix} \begin{pmatrix} R \\ G \\ B \end{pmatrix} = T_{YIQ} \begin{pmatrix} R \\ G \\ B \end{pmatrix} \tag{44}$$

Note that the $Y$ component is exactly the greyscale version $\mathbf{x}_{\text{gs}}$ defined above. We transform YIQ back to RGB by

$$\begin{pmatrix} R \\ G \\ B \end{pmatrix} = \begin{pmatrix} 1 & 0.956 & 0.619 \\ 1 & -0.272 & -0.647 \\ 1 & -1.106 & 1.703 \end{pmatrix} \begin{pmatrix} Y \\ I \\ Q \end{pmatrix} = T_{RGB} \begin{pmatrix} Y \\ I \\ Q \end{pmatrix} \tag{45}$$

| Parameter | CIFAR | Spirograph |
|---|---|---|
| Encoder model | ResNet50 | ResNet18 |
| Training batch size | 512 | 512 |
| Training epochs | 1000 | 50 |
| Optimizer | LARS | LARS |
| Scheduler | Cosine annealing | Cosine annealing |
| Learning rate | 3 | 3 |
| Momentum | 0.9 | 0.9 |
| Temperature $\tau$ | 0.5 | 0.5 |

Table 4: Hyperparameters used for CIFAR-10, CIFAR-100 and Spirograph

In YIQ format, we can adjust hue of an image by $\theta = 2\pi\alpha_{\text{hue}}$ by multiplying with a rotation matrix

$$R_\theta = \begin{pmatrix} 1 & 0 & 0 \\ 0 & \cos\theta & -\sin\theta \\ 0 & \sin\theta & \cos\theta \end{pmatrix} \tag{46}$$

Therefore, our hue adjustment is given by

$$\mathbf{x}_{\text{hue}} = T_{RGB} R_{\alpha_{\text{hue}}} T_{YIQ} \mathbf{x} \tag{47}$$

where the matrices operate on the three colour channels of $\mathbf{x}$ and in parallel over all spatial dimensions. Each operation is followed by pointwise clipping of pixel values to the range $[0, 1]$.

## F.2 SET-UP

Our set-up is quite similar to the setup in Chen et al. (2020a) with two main differences: we treat colour distortions as a differentiable module while in Chen et al. (2020a) the transformation was performed in the preprocessing step, and we add the gradient penalty term in addition to the original loss in Chen et al. (2020a).

### F.2.1 TRANSFORMATIONS

First, for a batch $\mathbf{x}_1, ..., \mathbf{x}_K$ of inputs, we form a pair of $(\mathbf{x}_1, \mathbf{x}'_1), ..., (\mathbf{x}_K, \mathbf{x}'_K)$ by applying two random transformations: random resized crop and random horizontal flip for each input. We then apply our differentiable colour distortion function which is composed of random colour jitter with probability $p = 0.8$ and random greyscale with probability $p = 0.2$. (Colour jitter is the composition of adjusting brightness, adjusting contrast, adjusting saturation, adjusting hue in this order.) We sample $\boldsymbol{\alpha}$, the parameter that controls how strong the adjustment is for each image from the following distributions: brightness, contrast and saturation adjustment parameters from $U(0.6, 1.4)$ and hue adjustment parameter from $U(-0.1, 0.1)$. We call the resultant pairs $(\mathbf{x}_1, \mathbf{x}'_1), ..., (\mathbf{x}_K, \mathbf{x}'_K)$.

### F.2.2 CONTRASTIVE LEARNING

Similar to Chen et al. (2020a), we use the transformed $(\mathbf{x}_1, \mathbf{x}'_1), ..., (\mathbf{x}_K, \mathbf{x}'_K)$ as an input to an encoder to learn a pair of representations $(\mathbf{z}_1, \mathbf{z}'_1), ..., (\mathbf{z}_K, \mathbf{z}'_K)$. The final loss function that we use for training is equation 13. Table 4 shows all hyperparameters that were used for training. The small neural network $\boldsymbol{g}_\phi$ is a MLP with the two layers consisting of a fully connected linear map, ReLU activation and batch normalization. We use LARS optimizer (You et al., 2017) and apply cosine annealing (Loshchilov & Hutter, 2016) to the learning rate.

### F.2.3 GRADIENT REGULARIZATION

In this part, we explain our setup for calculating the gradient penalty as in equation 12. We sample a random vector $\mathbf{e}$ with independent Rademacher components and independently for each sample in the batch. We generate $L$ samples of $\boldsymbol{\alpha}$ for each element of the batch to compute the regularizer. Finally, we clip the penalty from above to prevent instability at the onset of training. In practice, this meant the gradient regularization was not enforced for about the first epoch of training. Table 5 shows hyperparameters that we used within gradient penalty calculation.

| Parameter | CIFAR-10 | Spirograph |
|---|---|---|
| $L$ | 100 | 100 |
| $\lambda$ | 0.1 | 0.01 |
| Clip value | 1 | 1000 |

Table 5: Hyperparameters for gradient penalty calculation

| Parameter | CIFAR-10 | Spirograph |
|---|---|---|
| Evaluation model | Linear classification | Linear regression |
| Evaluation loss | Cross entropy loss | Mean squared error |
| Weight decay | $10^{-5}$ | $10^{-8}$ |
| Optimization | L-BFGS 500 steps | L-BFGS 500 steps |

Table 6: Hyperparameters for model evaluation

### F.2.4    EVALUATION

We use our representations as features in linear classification and regression tasks. We train these linear models with L-BFGS with hyperparameters as shown in Table 6 on the training set and evaluate performance on the test set.

### F.2.5    MoCo v2

To empirically demonstrate that our ideas transfer to alternative base contrastive learning methods, we applied both gradient regularization and feature averaging to the MoCo v2 (Chen et al., 2020c) base set-up. We also explored two different ResNet (He et al., 2016) architectures. We closely followed the MoCo v2 implementation at `https://github.com/facebookresearch/moco`. As for SimCLR, we adapted the transformations to be a differentiable module. We also made adaptations for CIFAR-100 in an identical way as in our previous experiments. As in MoCo v2, we removed batch normalization in the projection head $\mathbf{g}_\phi$; we used SGD optimization with learning rate 0.06 for a batch size of 512, and used the MoCo parameters $K = 2048$ and $m = 0.99$ for ResNet18 and $K = 4096, m = 0.99$ for ResNet50. We did not conduct extensive hyperparameter sweeps, but we did investigate larger values of $K$ which did not lead to improved performance on CIFAR-100. (In particular, the original settings $K = 65536, m = 0.999$ appeared to perform less well on this dataset.) Other hyperparameters and settings were identical to Chen et al. (2020c). We did 3 independent runs with a ResNet18 and 2 runs with a ResNet50. We conducted linear classification evaluation with fixed representations in exactly the same way as for our other experiments. Feature averaging results used $M = 40$.

### F.2.6    COMPUTATIONAL COST

We found that gradient regularization increased the total time to train encoders by a factor of *at most 2*. For feature averaging at test time with a fixed dataset, the computation of features $\mathbf{z}^{(M)}$ is an $O(M)$ operation, whilst the training and testing of the linear classifier is $O(1)$. Training time remained by far the larger in all experiments by orders of magnitude.

### F.3    ADDITIONAL EXPERIMENTAL RESULTS

### F.3.1    COMPARISON WITH ENSEMBLING

Feature averaging is an approach that bears much similarity with ensembling. To experimentally compare these two approaches, we applied both approaches to encoders trained on CIFAR-10. To provide a suitable comparison with feature averaging using $\mathbf{z}^{(M)}$ we first a trained a linear classifier $p(y|\mathbf{z})$ using an $M$-fold augmented dataset of representations with a standard cross-entropy loss using L-BFGS optimization using the same weight decay as for feature averaging. For CIFAR-10, which has a training set of length 50000, the feature averaging classifier was trained using 50000 averaged representations, whereas the ensembling classifier was trained with $50000M$ examples using data augmentation. At test time, we averaged prediction probabilities using $M$ different rep-

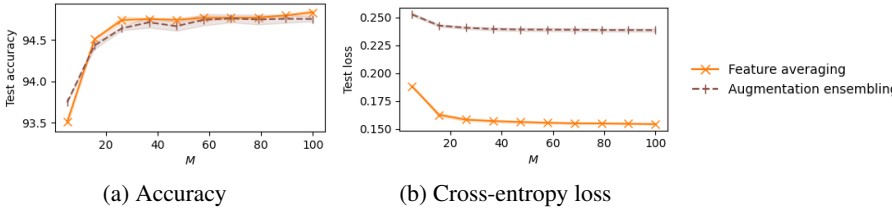

| (a) Accuracy | (b) Cross-entropy loss |

Figure 8: A comparison between feature averaging and augmentation ensembling using representations obtained with gradient regularization on CIFAR-10. Error bars are 1 s.e. from 3 runs.

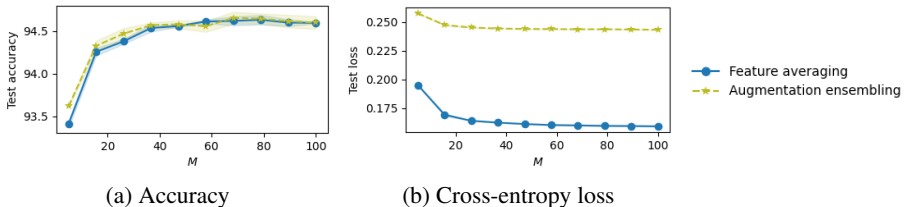

| (a) Accuracy | (b) Cross-entropy loss |

Figure 9: A comparison between feature averaging and augmentation ensembling using representations obtained with SimCLR on CIFAR-10. Error bars are 1 s.e. from 3 runs.

resentations of each test image. Specifically, if $p(y|\mathbf{z})$ is the classifier trained by the aforementioned procedure and $\boldsymbol{\alpha}_1, ..., \boldsymbol{\alpha}_M \sim p(\boldsymbol{\alpha}), \boldsymbol{\beta}_1, ..., \boldsymbol{\beta}_M \sim p(\boldsymbol{\beta})$ are independent transformation parameters, the probability of assigning label $y$ to input $\mathbf{x}$ is given by

$$p_{\text{ensemble}}(y|\mathbf{x}) = \frac{1}{M} \sum_{m=1}^{M} p(y|\mathbf{f}_\theta(\mathbf{t}(\mathbf{x}, \boldsymbol{\alpha}_m, \boldsymbol{\beta}_m))). \qquad (48)$$

The results outlined in Figure 8 show that ensembling gives very similar performance to feature averaging in terms of accuracy, but is significantly worse in terms of loss. We can understand this result intuitively because ensembling includes probabilities from every transformed version of the input (including where the classifier is uncertain or incorrect) whereas feature averaging combines transformations in representation space and uses only one forward pass of the classifier. More formally, the difference in test loss makes sense in light of Theorem 1. Figure 9 shows additional results obtained using representations trained with standard SimCLR on CIFAR-10. We see the same pattern—a similar test accuracy but worse test loss when using augmentation ensembling.

### F.3.2 GRADIENT REGULARIZATION LEADS TO STRONGLY INVARIANT REPRESENTATIONS

We first show that our gradient penalty successfully learns representations that have greater invariance to transformation than their counterparts generated by contrastive learning. We consider two metrics: the conditional variance targetted directly by the gradient regularizer, and the loss when $\mathbf{z}$ is used to predict $\boldsymbol{\alpha}$ with linear regression. Table 7 and Figure 10 are the equivalents of Table 1 and Figure 2 for CIFAR-100, showing the conditional variance and the regression loss for predicting $\boldsymbol{\alpha}$ respectively. In Table 8 we present the same results for Spirograph. We see very similar results to CIFAR-10 in both cases—the gradient penalty dramatically reduces conditional variances, and prediction of $\boldsymbol{\alpha}$ by linear regression gives a

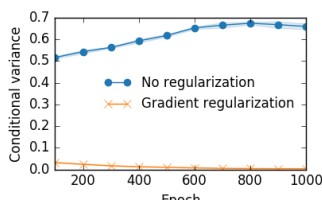

Figure 10: The conditional variance of Equation 6 on CIFAR-100. Error bars represent 1 s.e. from 3 runs.

loss that is better than a constant prediction only for standard contrastive representations.

### F.3.3 GRADIENT REGULARIZED REPRESENTATIONS PERFORM BETTER ON DOWNSTREAM TASKS AND ARE ROBUST TO TEST TIME TRANSFORMATION

For downstream performance on Spirograph, we evaluate the performance of encoders trained with gradient regularization and without gradient regularization on the task of predicting the generative

Table 7: The test loss when a linear regression model is used to predict $\alpha$ from $\mathbf{z}$ on CIFAR-100. The reference value is $\text{Mean}_i\,\text{Var}(\alpha_i)$. We present the mean and $\pm 1$ s.e. from 3 runs.

|  | Test loss |
| --- | --- |
| No regularization | $0.0357 \pm 0.0003$ |
| Regularization | $0.0417 \pm 0.00007$ |
| Reference value | $0.0408$ |

Table 8: Invariance metrics for Spirograph. We present the conditional variance, and the test loss when a linear regression model is used to predict $\alpha$ from $\mathbf{z}$. The reference value is $\text{Mean}_i\,\text{Var}(\alpha_i)$. We present the mean and $\pm 1$ s.e. from 3 runs.

|  | Conditional variance | Test loss |
| --- | --- | --- |
| No regularization | $0.789 \pm 0.0069$ | $0.0751 \pm 0.0003$ |
| Regularization | $0.0016 \pm 0.00004$ | $0.0808 \pm 0.00009$ |
| Reference value | - | $0.0806$ |

parameters of interest. In our set-up, we use 100,000 train images and 20,000 test images and train the encoders on the training set for 50 epochs. For evaluation, we train a linear regressor on the representations from encoders to predict the actual generative parameters. Setting for the linear regressor is shown in the Table 6. To accompany the main results in Figure 3(c), we include the exact values used in this figure in Table 9.

We now turn to our experiments used to investigate robustness—we investigate scenarios when we change the distribution of transformation parameters $\alpha$ at test time, but use encoders that were trained with the original distribution. We investigate on both CIFAR and Spirograph datasets.

For CIFAR, we chose to vary the distribution of parameters for colour distortions at test time. We could write the distribution of parameter of brightness, saturation, contrast as $U(1 - 0.8S, 1 + 0.8S)$ and the distribution of hue as $U(-0.2S, 0.2S)$ where $S$ is a parameter controlling the strength of the distortion. In the original setup, we have $S = 0.5$. By varying the value of $S$ used at test time, we can increase the variance of the nuisance transformations, including stronger transformations than those that were present when encoders were trained. This is visualized in Figure 11. Figure 14 is a companion plot for Figure 4(a) applied on CIFAR-100. We see broadly similar trends—our representations outperform those from standard contrastive learning across a range of test time distributions.

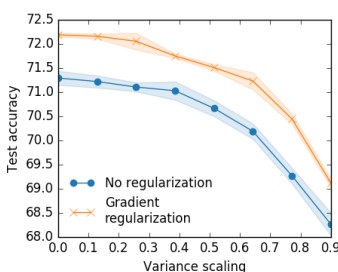

Figure 14: Robustness of performance on CIFAR-100 under variance scaling of transformation parameters.

For robustness on Spirograph recall $\alpha = (h, f_g, f_b, b_r, b_g, b_b)$ when $h \sim U(0.5, 2.5)$, $f_g, f_b \sim U(0.4, 1)$, $b_r, b_g, b_b \sim U(0, 0.6)$. We chose to vary the distribution of the background colour $(b_r, b_g, b_b)$ and the distribution of $h$ which is a structure-related transformation parameter. We consider two approaches, *mean shifting* where we shift the uniform distribution by $S$, for example $U(a, b) \rightarrow U(a + S, b + S)$ and we consider *changing variance* where we increase the range of the

Table 9: The raw values used to produce Figure 3(c). Each of the downstream tasks is a generative parameters. Values are the test mean square error $\pm 1$ s.e. from 3 runs.

| Parameters | No regularization | Gradient regularization |
| --- | --- | --- |
| $m$ | $0.0006773 \pm 0.0000509$ | $\mathbf{0.0005073 \pm 0.0000078}$ |
| $b$ | $0.0112480 \pm 0.0002555$ | $\mathbf{0.0073607 \pm 0.0001079}$ |
| $\sigma$ | $0.0000914 \pm 0.0000108$ | $\mathbf{0.0000527 \pm 0.0000026}$ |
| $f_r$ | $0.0000232 \pm 0.0000004$ | $\mathbf{0.0000028 \pm 0.0000001}$ |

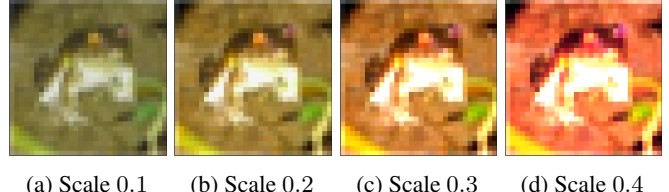

(a) Scale 0.1      (b) Scale 0.2      (c) Scale 0.3      (d) Scale 0.4

Figure 11: Visualization of test time distortions applied to CIFAR-10 for various variance scalings.

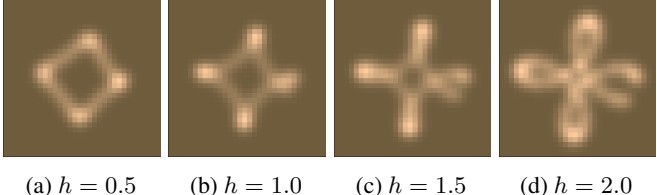

(a) $h = 0.5$      (b) $h = 1.0$      (c) $h = 1.5$      (d) $h = 2.0$

Figure 12: The effect of varying $h$ on the structure of a Spirograph image.

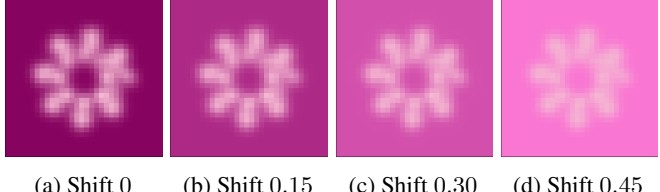

(a) Shift 0      (b) Shift 0.15      (c) Shift 0.30      (d) Shift 0.45

Figure 13: The effect of shifting the background colour distribution used for Spirograph images.

uniform distribution by $2S$, for example, $U(a,b) \rightarrow U(a - S, b + S)$. We compare performance of the trained encoders at epoch 50 on predicting the generative parameters $(m, b, \sigma, f_r)$ and we use the same setting for linear regressors as in Table 6.

Figure 12 is a visualization of the effect of varying $h$ from $0.5$ to $2.0$ while other parameters are kept constant. The figure 13 shows the effect of varying the background colour of an image by adding $S = 0.15, 0.30, 0.45$ to each of the background RGB channels.

For varying the distribution of $h$, we consider shifting the mean of $h \sim U(0.5, 2.5)$ by $S = \pm 0.1, \pm 0.3, \pm 0.5$ and increasing the variance of $h$ by $S = 0.1, 0.3, 0.5$. For the distribution of the background colour $(b_r, b_g, b_b)$, we consider shifting the distribution of $(b_r, b_g, b_b)$ by $S = 0.1, 0.2, 0.3, 0.4$ and increasing variance by the same amount. We note that $(b_r, b_g, b_b)$ controls the background colour of an image, so we are varying the 3 distributions at the same time. Since, the foreground colour has the distribution $f_r, f_g, f_b \sim U(0.4, 1)$, we are shifting the distribution of $(b_r, b_g, b_b)$ toward $(f_r, f_g, f_b)$ and this will make the background and foreground colours more similar. For example, with $S = 0.4$, when we apply a mean shift we change the distribution of $(b_r, b_g, b_b)$ to $b_r, b_g, b_b \sim U(0.4, 1)$, and when we increase the variance the distribution becomes $b_r, b_g, b_b \sim U(0, 1)$.

