# OpenReview forum: "Improving Transformation Invariance in Contrastive Representation Learning"
_ICLR.cc/2021/Conference — ICLR 2021 Poster_

### Official Review · AnonReviewer4 · 2020-10-26
**More direct enforcement of invariance, with an effective test time strategy**

**Rating:** 7
**Confidence:** 4

**Review:**

The paper focuses on a subtle different yet seemingly novel approach to learn invariance, i.e. instead of forcing the representations of two transformed inputs to be similar, it minimizes the gradient of the representation w.r.t to the transformation. This is indeed closer if not exactly the definition of tolerance or invariance to transformations is.

The problem is certainly an important one. The paper uses trick to to use a different representation of the conditional variance and first order approximation to arrive at a more computationally viable formulation. The formulation overall is interesting and described in sufficient detail. The paper does provide some relations to adversarial robustness as well, indeed it is quite intuitive to find connections between robustness and invariance.

The paper also provides validation experiments to show that indeed the conditional variance is reduced using this regularization.  However, there seems to be missing plots depicting this difference as a function of lambda. Is the method highly sensitive to lambda? or does a large range provide similar effects? If its the latter case, why so? what is preventing a large decrease in conditional variation given this loss augmentation?

Averaging across test samples does seem to help significantly. This is an important property in practice when there might be multiple transformations available that one needs to optimally utilize. This is one solution for such a case.

How would these ideas apply to more complicated nuisance transformations such as object pose?

---

> ### Author Response · Authors · 2020-11-20
> **Individual reviewer response**
>
> Thank you for your review. We are very glad that you think this work provides a novel approach to an important problem. We very much agree that the connection between robustness and invariance is an intuitive one. We have added several new experiments, including an investigation into the sensitivity with respect to the hyperparameter lambda. We hope the following more detailed responses addresses all your questions.
>
> > *There seems to be missing plots depicting this difference as a function of lambda. Is the method highly sensitive to lambda? or does a large range provide similar effects? If it's the latter case, why so?*
>
> Thanks for this great suggestion. We have duly added an additional ablation study [Figure 6] that shows how the conditional variance and the downstream task performance change as lambda is varied. As expected, the conditional variance decreases as lambda increases. The downstream task performance improves, before starting to get worse as lambda is made too large. We found a broad range of lambda values gave good performance: everything between 1e-4 and 1e-2 worked well. This corresponds to a fairly wide spectrum of suitable values.  We believe that the reason for this is that being effective requires a regularizer that can be brought close to 0 during the course of training but does not dominate over the contrastive term, a relatively weak requirement.
>
> > *What is preventing a large decrease in conditional variation given this loss augmentation?*
>
> The contrastive term in the loss (cf. equation 13) may prevent a large decrease in conditional variance for very small lambda values. In particular, the contrastive loss cannot be small for “trivial” solutions to minimizing the conditional variance, e.g. mapping every input to the same representation. Thus, our algorithm performs well when the two terms of equation 13 are well-balanced. In practice, the conditional variance does decrease to a small value when regularization is used, but the contrastive loss is allowed to simultaneously decrease so this is not problematic.
>
> >*How would these ideas apply to more complicated nuisance transformations such as object pose?*
>
> This is an interesting question.  As a continuous transformation parameter, our approach would naturally apply to object pose. The pose transformation variable (e.g. angles describing pose) would be a part of the transformation parameter alpha. We could then apply our gradient regularization to ensure that representations change slowly as pose is changed. At test time, information from multiple poses could be combined using feature averaging. A more complex relationship between pose and representation (e.g. equivariance) would be an exciting topic for future work.

---

### Official Review · AnonReviewer2 · 2020-10-28
**Reasonable idea, while some of the contributions are not novel enough against existing works.**

**Rating:** 6
**Confidence:** 4

**Review:**

1. Summary
Given one image, the paper first generates different views which are controlled by differentiable parameter \alpha, and then minimizes the additional "conditional variance" term~(expectation of these views' squared differences). Therefore, the paper encourages representations of the same image remain similar under the augmentation. A testing strategy is further proposed by voting features with different augmentations. Results demonstrate the effectiveness.

2. Strength
    * The proposed "conditional variance" and its first-order approximation which is driven by a learnable augmentation control factor \alpha is a clear contribution，and makes it efficient to minimize the representations of different views of the same image.
    * Results in Cifar and the new proposed dataset validate the effectiveness of the methods.

3. Weakness
    * Correct me if I were wrong. It seems more than half of the improvements come from the testing strategies. The claimed contribution of "feature averaging" during testing looks more like a testing time augmentation for me, although the paper has some differences in the augmentation details. I also think that the claim of SOTA is not rigorous if it has benefited from testing strategies.
    * It seems the proposed methods can be plugged into many recent studies. Thus more comprehensive experiments are needed to validate its generalization, e.g., combining with other baselines and conducting on ImageNet.

---------------------------
After reading the authors’ feedback and comments from other reviewers, I raised my score from 5 to 6.  I agree the feature averaging is another testing strategy.

---

> ### Author Response · Authors · 2020-11-20
> **Individual reviewer response**
>
> Thank you for your review. We are very pleased that you think our work constitutes a clear contribution and that our results demonstrate effectiveness. We agree that our method can be plugged into many recent methods, and feel that this is a further strength of our ideas. Following the suggestions made by yourself and other reviewers, we have added a significant number of new experiments to the paper, including an experiment with MoCo v2 as the base method in place of SimCLR. We believe the new experiments substantially strengthen the work.  We hope that you agree and ask you to consider increasing your score if you do.
>
> We now present detailed answers to your questions.
>
> > *It seems the proposed methods can be plugged into many recent studies. Thus more comprehensive experiments are needed to validate its generalization, e.g., combining with other baselines and conducting on ImageNet.*
>
> We agree that a great benefit of our method is that it can be used with a wide variety of base methods and thank you for the sensible suggestion of adding such experiments. We have duly carried this out by conducting a new experiment using MoCo v2 as the base method on the CIFAR-100 dataset [Table 3]. We see that our approach again leads to significant improvements in performance over the baseline approach: for ResNet50 we achieved more than a 6% improvement.
>
> Training with ImageNet is also a great idea, but we are unfortunately limited in our computational resources such that this will take us some time.  As a first step, we have performed an experiment using pre-trained MoCo v2 encoders and showed that feature averaging with M=20 improved top-1 accuracy on ImageNet from 71.1% to 71.9%, confirming that our approaches can also offer improvements on large scale datasets. We have started training MoCo v2 from scratch with our full approach for ImageNet, this will take us a few weeks to run so will not be ready before the end of the rebuttal period, but we will add the new results when it is done.
>
> > *Correct me if I were wrong.  It seems more than half of the improvements come from the testing strategies… I also think that the claim of SOTA is not rigorous if it has benefited from testing strategies.*
>
> We feel there may have been a misunderstanding here: our approach does not perform any model updates or fine tuning at test time; we use linear evaluation on frozen representations, in line with standard practice in the literature.  Thus, though we agree that feature averaging leads to improved performance, we disagree that this means our results are not rigorous in any way, or that they form inappropriate comparisons to previous SOTA.
>
> To be more precise, our methodology at test time creates a single representation for each training and testing image without changing the encoder. Using these frozen representations, a linear classifier is trained on the representation training set, which is then evaluated on the representation test set. This aligns with standard practices in the literature e.g. SimCLR, MoCo, etc.
>
> To further contrast our approach with other established methodologies: unlike test time augmentation, our linear classifiers are not trained on a longer dataset (for CIFAR, our linear classifier is always trained on a dataset of length 50000 of averaged representations); unlike augmentation ensembling, predictions from the linear classifier only require one pass through the classifier; unlike fine tuning, gradients do not flow to the encoder network at test time.
>
> We strongly believe our results are rigorous and comparable to other approaches that fit a linear classifier to frozen features.  In fact, by following careful statistical good practices, such as using repeat runs to test statistical significance, we feel that our paper is unusually rigorous for the subfield (as noted by another reviewer).
>
> > *The claimed contribution of "feature averaging" during testing looks more like a testing time augmentation for me*
>
> As noted by reviewer 1, feature averaging and test time augmentation are related but distinct ideas.
>
> We have introduced a new experiment that compares feature averaging with test time augmentation and augmentation ensembling. See Figure 7 for the results from this experiment and Appendix F.2.5 for full details of the distinction between feature averaging and augmentation.
>
> To explain the distinction in more detail: if using augmentation to train the linear classifier, one would use an expanded dataset with the representations of multiple differently augmented versions of the same input considered separately. In our feature averaging approach, the dataset to train the linear classifier is of a fixed size, but the representations are formed as an average of multiple features. At testing time, with augmentation ensembling, the prediction probabilities would be averaged as in Equation 51, whilst we instead perform a single forward pass through the linear model with a feature averaged representation.

---

### Official Review · AnonReviewer1 · 2020-11-03
**a very well done paper proposing methods to improve transformation invariance in contrastive learning with strong empirical results.**

**Rating:** 7
**Confidence:** 4

**Review:**

Summary:

- Proposes new method to improve transformation invariance in contrastive representation learning and demonstrates utility on downstream tasks
- Proposes using feature averaging from multiple transformations at test time leading to further improvements
- Introduces Spirograph dataset to explore the importance of learning feature invariances in the context of contrastive learning

Clarity
- Paper very well written and easy to follow. Figures supplement the text well
- Experiments include error bars to show statistical significance of results
- Supplementary material clarifies experimental setup and very comprehensive
- Perhaps consider changing “Self-supervized” -> “self-supervised”?

Novelty/Significance
- New contrastive objective with gradient regularizer term to encourage transformation invariance and the Spirograph dataset are well motivated
- Results over contrastive baselines suggest the contributions are important improvements to the contrastive training recipe

Questions/Comments/Clarifications
- “Unfortunately, directly changing the similarity measure hampers the algorithm.” - Please add a citation to validate this claim. Is it coming from the experiments in SimCLRv1 - Chen et al, 202

- “However, there are many ways to maximize the InfoNCE objective without encouraging strong invariance in the encoder.” - Please add a citation to validate this claim

- Eq 9 and 10:  (Fe(α, β, x) − Fe(α’ , t, x))^2. Should t not be β?

- “It may be beneficial, however, to aggregate information from differently transformed versions of inputs to enforce invariance more directly” -> it is unclear why invariances need to enforced at test time if the learned representation is already invariant

- The authors point this out but the gradient regularization term is unfortunately encouraging invariance only to differentiable
 transforms and this is a key limitation

- Worth pointing out for certain tasks likely near OOD detection, you may want to be transformation covariant rather than invariant. Would be interesting to see results of the proposed method on OOD detection benchmarks following https://arxiv.org/abs/2007.05566
- One major limitation is lack of baselines beyond vanilla contrastive training. For example, it would have been good to compare test time feature averaging with test time augmentation ensembling. Similarly instead of gradient regularization, the model could directly predict augmentation parameters and have the gradients of that loss penalized/ reversed. Adding such additional baselines, would solidify the improvements as best in class.
- It is also unclear how much train/test time compute model adds.

Overall, this is a very nicely written paper and very solid contribution. If the authors address my concerns, would be happy to increase my score.

---

> ### Author Response · Authors · 2020-11-20
> **Individual reviewer response**
>
> Thank you for a thoughtful review. We are really pleased that you found our work to be a well-written, novel contribution, with results indicating that our ideas are important improvements to the contrastive learning recipe. We’re also glad that the comprehensive supplement and error bars on every figure were valued. You mentioned a lack of baselines as limitation, and we have now included several new experiments which we hope directly address this point. Thank you for picking up typos which should be fixed in the updated paper. We hope the following more detailed discussion addresses the questions you had.
>
>
> > *“Unfortunately, directly changing the similarity measure hampers the algorithm.” - Please add a citation to validate this claim. Is it coming from the experiments in SimCLRv1 - Chen et al, 2020*
>
> We have added citations for this point.
>
> > *“However, there are many ways to maximize the InfoNCE objective without encouraging strong invariance in the encoder.” - Please add a citation to validate this claim*
>
> We have added a footnote with additional explanation and citation for this point.
>
> > *It is unclear why invariances need to enforced at test time if the learned representation is already invariant*
>
> There are two key reasons for this is still needed: 1) the training is typically not perfect such that further forcing of invariance is beneficial, and 2) gradient regularization only induces invariance in the differentiable transforms.
>
>
> > *Worth pointing out for certain tasks likely near OOD detection, you may want to be transformation covariant rather than invariant.*
>
> This is a great point and we have added a note on the subject in Appendix D; it could be a very interesting avenue for future research.
>
> > *One major limitation is lack of baselines beyond vanilla contrastive training. For example, it would have been good to compare test time feature averaging with test time augmentation ensembling*
>
> Thank you for this suggestion, we agree that it is a good way to strengthen the paper further. To this end, we have added a new experiment that compares feature averaging with augmentations ensembling (Sec 6.7 and Figure 7). We found that, whilst ensembling does nearly as well as feature averaging in terms of accuracy, the test loss is far worse. This result makes sense in light of Theorem 1.
>
> We have also added additional comparisons by considering the effect of applying our approaches in the context of the MoCo v2 model, finding that they offer substantial improvements here as well (see Table 3).
>
> > *The model could directly predict augmentation parameters and have the gradients of that loss penalized/ reversed*
>
> An adversarial training baseline would also be interesting although we did not have time to implement it during the rebuttal phase. We will aim to add this into the final version of the paper nonetheless.
>
> > *It is also unclear how much train/test time compute model adds*
>
> We have added a discussion of computational cost in Section F.2.7. In summary, the addition of gradient regularization typically increased training times by a factor of at most 2. Computing feature averaged representations scales as O(M), whilst training and testing the linear classifiers themselves is O(1). Training time remained by far the larger in all experiments by orders of magnitude.

---

### Author Response · Authors · 2020-11-19
**Summary response to all reviewers with additional experiments**

We want to thank all the reviewers for their efforts and the thoughtful and attentive examination of our work that they have offered.  We were glad to see that the reviewers highlighted a number of strengths including

 - The problem tackled by this work is “is certainly an important one” (R4)
 - The proposed gradient regularization is “a clear contribution” (R2) that is intuitive and “well motivated” (R1) and constitutes a “novel approach to learn invariance” (R4)
 - The proposal of feature averaging gives an “important property in practice” (R4) that leads to improved performance (R1, R2, R4)
 - The paper is “very well written and easy to follow” (R1) and “described in sufficient detail” (R4)
 - The experimental results already presented “suggest the contributions are important” (R1) and “demonstrate the effectiveness” (R2) of the approach.
 - The figures and supplementary material are comprehensive and complement the main text (R1)
 - The methods can be “plugged into many recent studies” and generalize to a range of methods (R2)

The main concerns raised revolved around a desire to see some additional experimental results, for which a number of sensible suggestions were made for improvements. To this end, we have now run a large number of new experiments and added these to the paper in the new revised version, namely we have:
 - Shown that our approach can be successfully applied to other base models, by combining it with MoCo V2 on CIFAR-100 [Table 3]
 - Performed an ablation study for the key hyperparameter, lambda, by examining the effect of varying it on test loss and conditional variance for the spirograph dataset [Figure 6], finding that there is a stable region of effective values. We will include a detailed discussion about what prevents a large decrease in conditional variance in the individual response.
 - Added an additional baseline by comparing test feature averaging with test time augmentation ensembling on CIFAR-100 [Figure 7], finding that our approach produces lower test losses.

We thank the reviewers for their suggestions for additional experiments that enhance this paper. We feel that this substantial new set of experiments significantly strengthens the work and hope the reviewers agree.

We would also like to thank reviewers for picking up typos and missing references which should be corrected in the updated version of the paper.

Reviewers also had a number of specific questions that we will answer with individual responses. We are still working on these responses to individual reviewers, but hope to have these with you tomorrow. We will also add appendices providing full details of the new experiments.

Thank you all again for your hard work and consideration!

---

### Decision · Program_Chairs · 2021-01-07
**Final Decision**

**Decision:**

Accept (Poster)

**Comment:**

The paper recieved three consistently positive reviews. While I agree with most of them, I have two major concerns regarding the novelty of the paper, which the authors are strongly recommended to address in the final version.

1. Taking derivative with respect to the parameters of transformation isn't novel. The standard tangent prop algorithm has been around for over a decade:

P. Simard, Y. LeCun, J. S. Denker, and B. Victorri. Transformation invariance in pattern recognition-tangent distance and tangent propagation. In Neural Networks: Tricks of the Trade. 1996.
(see Eq 26 in https://halshs.archives-ouvertes.fr/halshs-00009505/document)

Salah Rifai, Yann N. Dauphin, Pascal Vincent, Yoshua Bengio, Xavier Muller. The Manifold Tangent Classifier. NIPS 2011.
(see Eq 6 therein)

I understand there is some normalization in Eq 5, sampling of \alpha, \alpha', and the direction, and using the expectation to approximate the norm of the gradient. But such novelty is really incremental, or at least some empirical comparison will be necessary. It will also be necessary to cite the tangent distance/prop literature.

2. The new gradient based regularizer in Eq 11 and 12 appears completely decoupled with contrastive learning. It can be applied to any representation learning where f_\theta is an encoder. It does not use any substantial element from contrastive learning, although it might be "inspired" by contrastive learning. One may argue that such generality is an advantage, but 1) there is really no need to take such a big detour into contrastive learning just in order to derive the invariance regularizer in Eq 12, and 2) writing in this way can be quite confusing and/or misleading.

---

> ### Author Response · Authors · 2021-03-22
> **Thank you and clarifying remarks**
>
> We are very pleased that our work has been accepted by ICLR 2021. Thank you very much for your role in this and also for your helpful additional comments; we have made edits for the final paper based on them and provide some additional clarifying discussion below for the record.
>
> Regarding the connection with earlier work, we have added the citations that you suggested along with a discussion on the connections between our work and existing methods such as tangent prop. We do believe that our approach has significant differences from this earlier work. Using the notation of this paper, tangent propagation penalizes the norm of the gradient of a neural network evaluated at α = 0, encouraging local transformation invariance near the original input. In our work, we target the conditional variance (Equation 6), leading to gradient evaluations across the α parameter space with random α ∼ p(α) and a regularizer that is not a gradient norm (Equation 12).
>
> We agree that our gradient regularization could be applied in other representation learning settings, and this may actually be a virtue of our ideas. The key focus of this paper was to understand the role of transformation invariance in contrastive learning, in many ways seeking to answer the question “How relevant is invariance to contrastive representation learning?” The approach we took to answering this was to enforce stronger invariance in contrastive learning and show that this results in improved performance on downstream tasks. This is in contrast with e.g. tighter mutual information bounds, which do not always translate into improved performance on downstream tasks. Finally, as we discuss in Section 2, transformation invariance is intrinsically a part of contrastive learning in a way that it is not intrinsically a part of e.g. representation learning with VAEs.